# Flow Autoencoders are Effective Protein Tokenizers

**Rohit Dilip**
California Institute of Technology
rdilip@caltech.edu

**Evan Zhang**[*][†]
OpenAI
evanz@openai.com

**Ayush Varshney**[*]
Carnegie Mellon University
varshney@cmu.edu

**David Van Valen**
California Institute of Technology
Howard Hughes Medical Institute
vanvalen@caltech.edu

## Abstract

Protein structure tokenizers enable the creation of multimodal models of protein structure, sequence, and function. Current approaches to protein structure tokenization rely on bespoke components that are invariant to spatial symmetries, but that are challenging to optimize and scale. We present Kanzi, a flow-based tokenizer for tokenization and generation of protein structures. Kanzi consists of an autoencoder trained with a flow matching loss. We show that this approach simplifies several aspects of protein structure tokenizers: frame-based representations can be replaced with global coordinates, complex losses are replaced with a single flow matching loss, and SE(3)-invariant attention operations can be replaced with standard attention. We find that these changes stabilize the training of parameter-efficient models that outperform existing tokenizers on reconstruction metrics at a fraction of the model size and training cost. An autoregressive model trained with Kanzi outperforms similar generative models that operate over tokens, although it does not yet match the performance of state-of-the-art continuous diffusion models.

## 1 Introduction

The promise of digital biology is to develop machine learning models that are capable of performing a wide range of tasks, from generating novel therapeutics to reasoning about cellular-level processes (Richardson & Richardson, 1989; Cui et al., 2025; Kuhlman & Bradley, 2019). Proteins, which are essential components of almost all biological processes, are a natural target for machine learning approaches to biological perception and generation. A recent exciting advance has been the development of multimodal deep learning models capable of reasoning over protein sequence, structure, and function (Wang et al., 2024; Zhang et al., 2024; Liu et al., 2024; Gaujac et al., 2024; Hayes et al., 2025). These models are enabled by structure tokenizers, which convert the continuous three-dimensional protein structures into a sequence of discrete tokens from a finite vocabulary using vector quantization (Van Den Oord et al., 2017). In these tokenizers, protein structures are represented as tensors $\mathbb{R}^{L \times A \times 3}$, where $L$ is the sequence length and $A$ is the number of backbone atoms. Training language models on these discrete token sequences unlocks the possibility of truly multimodal biological models that excel across representation and generative tasks.

An established practice in protein tokenization is using model components that are invariant to spatial symmetries, namely SE(3) (the special Euclidean group, equivalent to $SO(3) \ltimes \mathbb{R}^3$). This follows the hypothesis that by explicitly encoding inductive biases, models will not hallucinate physically implausible samples that break the symmetry. These SE(3)-invariant modules, however, can be challenging to both optimize at scale and extend to more diverse biological molecules (e.g., proteins with

---

[*]Equal contribution.
[†]Work done while not at OpenAI.

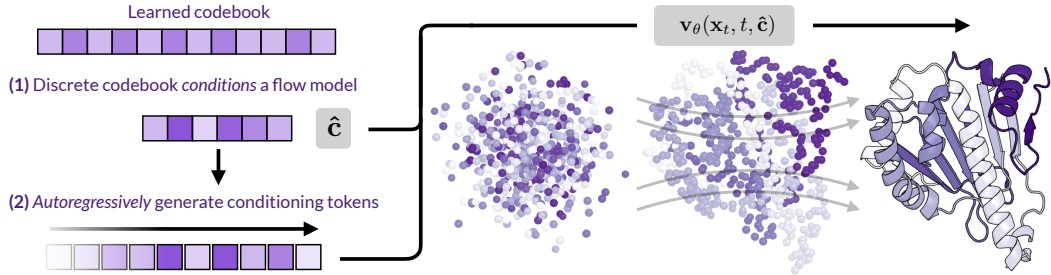

Figure 1: Schematic overview of our approach. (1) A learned discrete codebook conditions a flow matching model to reconstruct proteins using a diffusion loss. (2) The learned tokens can be used for downstream autoregressive generation, and the generated tokens condition the diffusion decoder to generate protein backbones.

post-translational modifications, RNA, DNA). Models that can accurately tokenize protein structures without explicitly encoding spatial symmetries may offer improved flexibility and scalability for modeling biology. Such models, however, do not currently exist.

Bridging this gap and exploring the performance of non-invariant protein tokenizers is the primary focus of this work. In this work, we describe Kanzi, a flow-based tokenizer for protein structures. Kanzi uses a flow matching loss to train an autoencoder that tokenizes protein structures. This flow loss simplifies model training by replacing the collection of symmetry-invariant reconstruction losses that are commonly used to train protein structure tokenizers. Kanzi operates directly on the 3D coordinates of backbone atoms and uses standard attention rather than SE(3)-invariant geometric attention methods (Jumper et al., 2021; Hayes et al., 2025). Consistent with several recent works that challenge the inductive bias paradigm (Abramson et al., 2024; Geffner et al., 2025), Kanzi demonstrates that these simplifications can actually improve tokenizer performance, achieving superior reconstruction quality over larger models trained on more extensive datasets. We next train an autoregressive model on tokenized structures from Kanzi to sample plausible protein structures. While discrete and continuous diffusion models have seen wide use, autoregressive models are better suited for generating variable-length sequences. This capability is critical for tasks like motif scaffolding or *in situ* structure prediction where the protein sequence (and hence length) is unknown a priori. Despite a relatively modest autoregressive model, we match or outperform larger tokenized models in terms of generation quality, as measured by standard benchmarks. As a supplementary contribution, we introduce a reconstruction metric, the reconstruction Fréchet Protein Structure Distance (rFPSD), which utilizes probability divergences to measure structure tokenization. This extends prior work in Faltings et al. (2025) and Geffner et al. (2025) that applies these metrics to generation. We provide an open-source software package as a standalone repository for end-users.

In summary, our primary contributions are as follows:

1. We present Kanzi, a scalable state-of-the-art **flow-based structure tokenizer** based on a novel asymmetric encoder-decoder design.

2. We demonstrate that a simple diffusion loss can replace complex invariant/equivariant tokenizer losses, yet still achieve SOTA reconstructions.

3. We use Kanzi to train an autoregressive protein structure generation model. On standard generative benchmarks, the resulting generations match or outperform existing generative capabilities. To our knowledge, this is the first tokenized model that produces designable structures without massive pretraining.

4. We extend prior work developing distributional metrics for proteins to the reconstruction task to provide broader information on tokenization performance. Through a series of careful ablations, we demonstrate that while non-invariant encoders can learn scalably, invariant encoders struggle to condition non-invariant decoders.

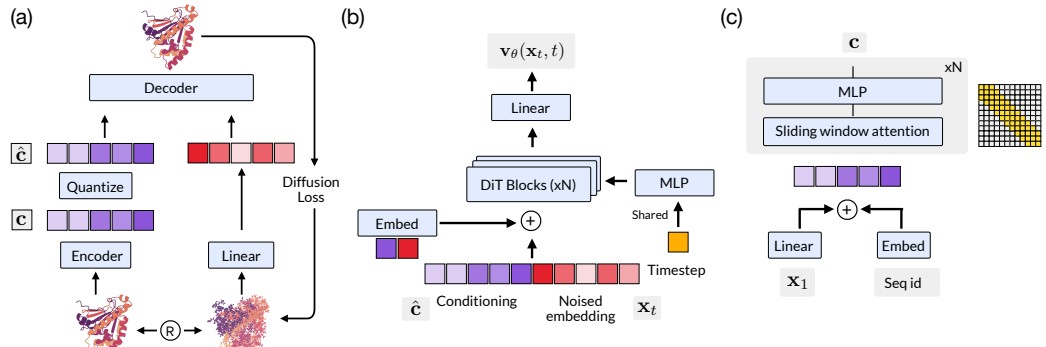

Figure 2: Architectural overview of Kanzi. (a) Kanzi takes a clean structure as input, which is encoded and passed through a quantization bottleneck. The decoder is provided with the quantized latents as in-context conditioning, along with a noised version of the protein structure. The training is supervised by a single diffusion loss that maximizes $p(\mathbf{x}|\hat{\mathbf{c}})$. No auxiliary losses are used. (b) Our decoder follows the standard diffusion transformer (DiT) presentation, with several notable deviations. We share adaLN conditioning across all blocks, and each DiT block is a transformer with pair-biased attention and optional self-conditioning. (c) Our encoder combines raw coordinate information with sequence positional information. Tokens are mixed using a small stack of transformer layers with sliding window attention. Ablations on other encoder variants are described in Section 4.3 and Appendix G.

## 2 RELATED WORK

**Tokenization.** State-of-the-art generative image models frequently first train an image *tokenizer*, which downsamples continuous image data to either a discrete or a continuous latent (Esser et al., 2021; Van Den Oord et al., 2017). Recent works in machine learning for biology have followed suit by training discrete tokenizers for protein backbone structures, which enable language models to be trained on sequences of tokens derived from the resulting codebooks (van Kempen et al., 2022; Steinegger & Söding, 2017; Gaujac et al., 2024; Lin et al., 2023). While tokenized protein models have underperformed diffusion models on the task of unconditional structure generation, they enable the construction of multimodal generative models of proteins. ESM3 notably trained a multimodal discrete diffusion model over sequence, structure, secondary structure, and natural language, which was capable of generating novel proteins with specified functions (Hayes et al., 2025; Wang et al., 2024). Following AlphaFold2, protein tokenizers generally rely on $SE(3)$-invariant architectural components (e.g., invariant point attention) and $SE(3)$-invariant losses (e.g., frame-aligned point error). In contrast with prior work, we use a non-invariant diffusion loss to supervise the tokenizer.

**Diffusion and flow matching.** State-of-the-art protein structure generation models rely on diffusion, either discrete or continuous. FrameDiff, RFDiffusion, and Chroma (Yim et al., 2023b; Watson et al., 2023; Ingraham et al., 2023) generate protein backbones using a denoising process over the joint translation-rotation group $SO(3) \ltimes \mathbb{R}^3$, while FrameFlow, FoldFlow, and FoldFlow-2 similarly perform flow matching over the same manifold (Yim et al., 2023a; Bose et al., 2023; Huguet et al., 2024b). Genie2 and Proteina are more recent attempts to train diffusion models at scale on the AlphaFold Structure Database (AFDB); these both operate over $C\alpha$ coordinates (Geffner et al., 2025; Lin et al., 2024). The latter does not explicitly encode invariances, a strategy we broadly adopt here. Despite the strong performance of diffusion models, autoregressive models have unique features that are valuable to the structural biology and machine learning communities. Most notably, they can be applied to more use cases where the protein size is not known *a priori*, an important feature for tasks such as motif scaffolding or *in situ* structure prediction in electron tomography images (Yadav et al., 2020; Bunne et al., 2024).

To train Kanzi, we use a flow matching objective (Esser et al., 2024; Lipman et al., 2022; Lin et al., 2024). Flow matching interpolates between a source distribution $p_0$ (often a Gaussian) and a target distribution $p_{\text{data}}$ by integrating along the ODE $d\mathbf{x}_t = \mathbf{v}_\theta(\mathbf{x}_t, t)\, dt$ using a learned vector field $\mathbf{v}_\theta(\mathbf{x}_t, t) : \mathbb{R}^d \times [0, 1] \to \mathbb{R}^d$. As the vector field generating the true probability distribution is in

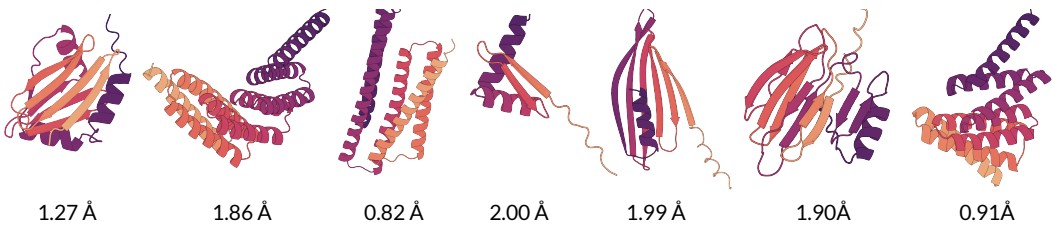

| 1.27 Å | 1.86 Å | 0.82 Å | 2.00 Å | 1.99 Å | 1.90Å | 0.91Å |

Figure 3: Designable samples generated from an autoregressive model trained on Kanzi tokens. scRMSDs shown underneath each visualization.

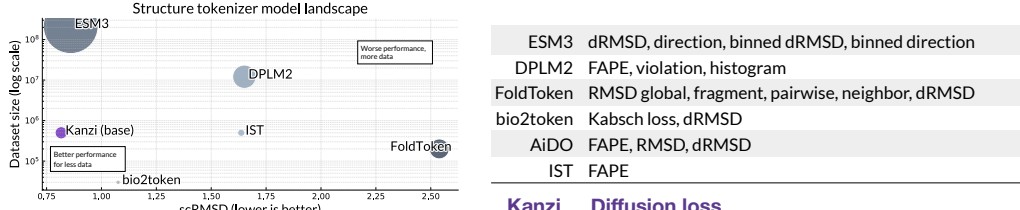

Figure 4: Left: Scaling of protein structure tokenizer performance with dataset size and parameter count. We plot the reconstruction accuracy on the CAMEO test set versus the training dataset size. Circle area is the model parameter count. Kanzi is competitive with the ESM3 tokenizer, despite a 20-fold smaller parameter count and 400-fold smaller training dataset. Right: Kanzi simplifies the training pipeline, replacing collections of complex, invariant losses with a single, non-invariant flow matching loss.

general unknown, one uses conditional flow matching, which constructs a conditional probability path between prior samples $\mathbf{x}_0 \sim p_0$ and data samples $\mathbf{x}_1 \sim p_{\text{data}}$. Explicitly, a general probability path can be written $\mathbf{x}_t = \alpha_t \mathbf{x}_1 + \sigma_t \epsilon$, with $\epsilon \sim \mathcal{N}(0, 1)$. This induces a true conditional vector field $\mathbf{u}(\mathbf{x}_t | \mathbf{x}_0, \mathbf{x}_1) = \dot{\alpha}_t \mathbf{x}_1 + \dot{\sigma}_t \mathbf{x}_0$, where the dot denotes the time derivative. This is a target we can regress against; for the standard case of the linear interpolation path, we have $\mathbf{x}_t = (1 - t)\mathbf{x}_0 + t\mathbf{x}_1$ and $\mathbf{u}(\mathbf{x}_t | \mathbf{x}_0, \mathbf{x}_1) = \mathbf{x}_1 - \mathbf{x}_0$. For completeness, we present a more thorough derivation of the flow matching formulation in Appendix H.

**Flow autoencoders.** The idea of using a flow or diffusion model as the decoder in tokenizer reconstruction is a recent insight in computer vision but has yet to be explored for protein structure generation (Preechakul et al., 2022). Two recent works, FlowMo and DiTo (Sargent et al., 2025; Chen et al., 2025), both study this approach and independently demonstrate SOTA performance on ImageNet-1k reconstruction. In these works, the use of a diffusion model eliminates the need for combinations of perceptual and adversarial losses during training, which were critical insights introduced by VQGAN (Esser et al., 2021). In our case, given the success of recent models like AlphaFold3, Boltz (Wohlwend et al., 2025), and Proteina in eschewing symmetric architectures for scalability, we hypothesize that flow autoencoders could provide a similar advantage for tokenization.

## 3 METHOD

We train flow-based tokenizers to perform autoregressive generation. The use of a diffusion/flow model as the decoder allows for considerably more flexibility and scalability in sampling, while the autoregressive prior trained over quantized sequences provides length-agnostic generation.

### 3.1 ARCHITECTURE

We design a non-equivariant flow autoencoder to tokenize an input structure. It consists of a lightweight encoder $e_\theta$, a substantially deeper decoder $d_\phi$, and a quantization bottleneck. A protein structure can be represented as a tensor $\mathbf{x} \in \mathbb{R}^{L \times A \times 3}$, where $L$ is the sequence length and $A$ is

the number of backbone atoms. Generally $A = 1$ for $C\alpha$ only models or $A = 3$ for full backbone models. Given $\mathbf{x}$, the encoder processes the raw, mean-centered coordinates using a stack of optionally pair-biased multi-head self-attention layers. The encoder outputs a latent conditioning sequence $e_\theta(\mathbf{x}) = \mathbf{c} \in \mathbb{R}^{L \times d}$.

For the quantization layer, we adopt finite scalar quantization (FSQ) and discretize $\mathbf{c}$ to a discrete latent $\hat{\mathbf{c}}$ via $\hat{\mathbf{c}} = \lfloor \ell/2 \rfloor \tanh(\text{Linear}(\mathbf{c}))$, where $\ell$ is the number of levels in each dimension of FSQ (Mentzer et al., 2023). We generally use 8,5,5,5 for an effective codebook size of 1000. We use $\hat{\mathbf{c}}$ to condition the diffusion decoder, which outputs $d_\phi(\mathbf{x}_t, t, \hat{\mathbf{c}}) = \mathbf{v}_\theta(\mathbf{x}_t, t, \hat{\mathbf{c}})$, with $\mathbf{x}_t$ the linearly interpolated noise. We pass gradients to the encoder using the standard straight-through estimator.

Figure 2 shows all the components of our architecture. First, in contrast with Sargent et al. (2025), which uses separate concatenated modality streams for both the encoder and the decoder, we use a *single* stream for the encoder but concatenated conditioning (i.e., two streams) for the decoder. Because our data is low-dimensional, we found that this was a critical design choice that allowed gradients to efficiently propagate through shallow encoders. In addition, the encoder uses sliding window attention, while the decoder has full bidirectional connectivity. This bias was included to facilitate autoregressive modeling; see Section G for more discussion. Otherwise, the encoder is significantly smaller than the decoder in both width and depth, which is a common design choice for tokenizers. We use relative positional encodings (RoPE) for the decoder, and ablate between absolute and relative positional encodings for the encoder. Finally, in contrast with the standard Diffusion Transformer (Peebles & Xie, 2023), we share adaLN weights across layers for time conditioning, which reduces the parameter count by $\approx 30\%$. We justify this choice in Appendix G. Our full hyperparameter selections are in Appendix F.1.

## 3.2 TRAINING

We optimize the entire tokenizer end-to-end using a flow loss

$$\mathcal{L}_{\text{flow}} = \mathbb{E}_{\mathbf{x}_1 \sim p_{\text{data}}, \, \mathbf{x}_0 \sim \mathcal{N}(0,1)} \|v_\theta(\mathbf{x}_t, t, \hat{\mathbf{c}}) - (\mathbf{x}_1 - \mathbf{x}_0)\|_2^2 \qquad \hat{\mathbf{c}} = \text{FSQ}\left(e_\theta(\mathbf{x})\right) \tag{1}$$

We again note the relative simplicity of this loss; Figure 4 and Appendix I.1 provide a detailed description of prior losses to supervise tokenization. We train until convergence using AdamW with $\beta_1 = 0.9, \beta_2 = 0.95$, and learning rate $\eta = 1.7 \times 10^{-4}$. We use a linear warmup and cosine decay schedule with random rotations on the inputs as augmentations. Details of our training/hyperparameter configurations are given in Appendix F.1. One advantage of flow autoencoders is the ability to use classifier-free guidance to improve sample quality. We mask out the conditioning sequence $\hat{\mathbf{c}}$ with probability $0.1$ to enable this option. Following AlphaFold3 and Proteina, we mean-center all proteins using $C\alpha$ coordinates and augment input structures with random rotations during training.

## 3.3 DATASET

While early protein generative models trained primarily on the $\sim 30k$ distinct structural homologs in the Protein Data Bank (Bank, 1971), recent works like Proteina and AlphaFold3 have trained on synthetic AlphaFold2 predictions to achieve significant performance gains. As Proteina and AlphaFold3 were trained on datasets that require extensive computational resources (Proteina trains on the full 214M structures in the AFDB, and AlphaFold3 trains on 40M MGnify structures), we instead train on the Foldseek clustered AlphaFold database, which we denote as $\mathcal{D}_{\text{FS}}$. $\mathcal{D}_{\text{FS}}$ filters and clusters the AFDB using MMseqs2 and Foldseek and keeps a single representative structure per cluster. Both Proteina and Genie2 include $\mathcal{D}_{\text{FS}}$ in their training data. We perform additional filtering (described in Appendix F.2) which leaves a total of 498,900 structures.

## 3.4 INFERENCE

**Diffusion decoding**. Given a sequence of Kanzitokens, to sample the full distribution we can simply run Euler inference using the standard integrator, i.e., we repeat the following for fixed $\hat{\mathbf{c}}$ and $t \in [0, 1/N, 2/N, ..., (N-1)/N]$.

$$\mathbf{x}_{t+\Delta t} = \mathbf{x}_t + \mathbf{v}_\theta(\mathbf{x}_t, t, \hat{\mathbf{c}})\Delta t \tag{2}$$

where $\hat{\mathbf{c}}$ is our conditioning sequence. We make several noteworthy additions. A major strength of flow tokenizers is their ability to take advantage of sophisticated sampling strategies at inference time compared to autoregressive or discrete diffusion samplers. We can construct new flows using classifier-free guidance with guidance parameter $g$ as follows:

$$\tilde{\mathbf{v}}_\theta(\mathbf{x}_t, t, \hat{\mathbf{c}}) = \mathbf{v}_\theta(\mathbf{x}_t, t, \hat{\mathbf{c}}) + g(\mathbf{v}_\theta(\mathbf{x}_t, t, \hat{\mathbf{c}}) - \mathbf{v}_\theta(\mathbf{x}_t, t, \varnothing)) \tag{3}$$

We omit the tilde for the remainder of the paper, with the understanding that we tune classifier-free guidance unless explicitly stated otherwise. As we use a Gaussian flow, we also have a closed-form expression for the corresponding score field

$$\mathbf{s}_\theta(\mathbf{x}_t, t, \hat{\mathbf{c}}) = \frac{t\mathbf{v}_\theta(\mathbf{x}_t, t, \hat{\mathbf{c}}) - \mathbf{x}_t}{1 - t} \tag{4}$$

An analogous expression for the score field with classifier-free guidance applies. A frequent practice in biological diffusion is to construct an ad hoc stochastic differential equation as an alternative sampler, as in Equation 5

$$d\mathbf{x}_t = \mathbf{v}_\theta(\mathbf{x}_t, t, \mathbf{c})\, dt + g(t)\eta\, \mathbf{s}_\theta(\mathbf{x}_t, t, \hat{\mathbf{c}})\, dt + \sqrt{2g(t)\gamma}\, d\mathcal{W}_t \tag{5}$$

This notation largely follows that of Geffner et al. (2025). Setting $\eta = \gamma = 1$ corresponds to standard Langevin dynamics. However, treating the noise scale $\gamma$ and the score scale $\eta$ as hyperparameters can significantly improve generation quality, although sometimes at the cost of diversity. In our benchmarks, we provide sampling using both the full distribution (e.g., $\eta = \gamma = 0$) and using noise and score-scaling as references.

**Autoregression.** We explored both nucleus sampling and min-p sampling for generating the conditioning sequence $\mathbf{c}$, but did not observe any significant difference between the two. All presented results use nucleus sampling with a cutoff of 0.9. As our generative models are autoregressive, we can also do best-of-N sampling with the log-likelihood as an inexpensive proxy for decoding quality, a strategy that has proven useful for large language models (Qiu et al., 2024; Song et al., 2024). Generally $N = 2$ or $4$ in our experiments.

## 4 EXPERIMENTS

We evaluate Kanzi on reconstruction, generative, and representative tasks. On reconstruction tasks, we find that flow autoencoders exceed or match the performance of much larger models trained on substantially more data. On generative tasks, we exceed or match the performance of larger tokenized models and consistently outperform comparably sized generative models trained on other tokenizers. We defer evaluations on representation quality to Appendix E; in summary, Kanzi outperforms similar non-invariant tokenizers on residue-level representation tasks but underperforms invariant tokenizers.

### 4.1 FLOW TOKENIZERS ARE SOTA AT RECONSTRUCTION

We first evaluate Kanzi on reconstruction. We benchmark reconstruction performance against all structure tokenizers with accessible public repositories: ESM-3, DPLM-2, Bio2Token, FoldToken, and the InstaDeep Structure Tokenizer (IST). A challenge with evaluating tokenizers is every model tends to use a different training/test set, which makes it challenging to compare model performance. To address this, we use a wide range of test datasets (all are held-out from our model, along with any structural homologs at 80% similarity as determined by Foldseek). We exclude any cases where we know there is leakage between the benchmark set and the tokenizer. See Appendix I.2 for more details on these determinations. We use five held-out test datasets: CAMEO, CASP14, CASP15, CATH, and a held-out subset of $\mathcal{D}_{\text{FS}}$. We use RMSD and TM-score to benchmark local and global measures of reconstruction, and include scores for both full backbone tokenizers and $C\alpha$ only tokenizers. We also introduce the following two auxiliary metrics.

**rFPSD** (reconstruction Fréchet Protein Structure Distance) provides a distribution-level metric for reconstruction by using the deep features in a pretrained CATH-classifier. Metrics like RMSD are

| | CAMEO | | CASP14 | | CASP15 | | CATH | | | | | AFDB | |
|---|---|---|---|---|---|---|---|---|---|---|---|---|---|
| | RMSD (↓) | TM (↑) | RMSD | TM | RMSD | TM | RMSD | TM | [$\beta$]RMSD | [$c$]RMSD | rFPSD | RMSD | TM |
| DPLM2 (118M) | 1.651 | 0.876 | 1.008 | 0.951 | 2.160 | 0.866 | 1.641 | 0.897 | 1.851 | 2.067 | **5.742** | 4.676 | 0.810 |
| ESM3 (648M) | 0.860 | 0.955 | **0.462** | **0.987** | **1.021** | 0.969 | 1.048 | **0.957** | 1.391 | 1.086 | 23.399 | 2.384 | 0.915 |
| FoldToken (85M) | 2.539 | 0.881 | 2.194 | 0.936 | 6.629 | 0.744 | 1.298 | 0.920 | 1.575 | 1.231 | 71.786 | 2.161 | 0.858 |
| IST (11M) | 1.637 | 0.916 | 0.900 | 0.960 | 1.252 | 0.953 | 1.201 | 0.940 | 1.127 | 1.246 | 105.208 | 2.872 | 0.862 |
| bio2token (1.1M) | 1.076 | 0.948 | 1.006 | 0.952 | 1.377 | 0.939 | - | - | 1.361 | 1.008 | - | 1.212 | 0.932 |
| Kanzi (30M) | 0.936 | 0.948 | 0.861 | 0.958 | 1.345 | 0.951 | 1.098 | 0.940 | 0.774 | 1.181 | 27.202 | 1.069 | 0.947 |
| Kanzi (30M)* | **0.817** | **0.960** | 0.698 | 0.972 | 1.267 | 0.963 | **0.953** | 0.955 | 0.658 | 1.023 | 7.956 | **0.870** | **0.962** |
| Kanzi (11M) | 1.016 | 0.937 | 0.912 | 0.954 | 1.259 | 0.955 | 1.156 | 0.934 | 0.805 | 1.239 | 20.104 | 1.210 | 0.934 |
| Kanzi (11M)* | 0.863 | 0.952 | 0.762 | 0.968 | 1.105 | 0.965 | 0.994 | 0.950 | 0.813 | 1.058 | 51.649 | 0.994 | 0.952 |

Table 1: Reconstruction metrics across tokenizers for $C\alpha$ reconstruction. Kanzi consistently matches or outperforms much larger models trained on larger datasets. Best result in **bold**, second best result underlined. Datasets like CAMEO and CASP are relatively small and have larger variances. We exclude any cases where a model is explicitly stated to be trained on a held-out dataset. Starred (*) Kanzi models use $\eta = 0.45, \gamma = 1.0, g = 2.0$. This parameter setting is deliberately underoptimized; we tuned against a small subset of our AFDB test set (100 structures) and applied it without adjustment to the held-out non-synthetic datasets. For visual clarity, standard errors are left to the Appendix.

| | CAMEO | | CASP14 | | CASP15 | | CATH | | AFDB | |
|---|---|---|---|---|---|---|---|---|---|---|
| | RMSD(↓) | TM (↑) | RMSD | TM | RMSD | TM | RMSD | TM | RMSD | TM |
| DPLM2 | 1.631 | 0.928 | 0.995 | 0.959 | 2.144 | 0.953 | 1.717 | 0.925 | 4.646 | 0.880 |
| ESM3 | **0.861** | **0.980** | **0.463** | **0.994** | **1.018** | **0.983** | 1.151 | 0.971 | 2.378 | 0.944 |
| FoldToken | 2.498 | 0.922 | 2.323 | 0.958 | 6.580 | 0.809 | 1.352 | 0.944 | 2.120 | 0.907 |
| IST | 1.626 | 0.954 | 0.896 | 0.974 | 1.244 | 0.975 | 1.195 | 0.956 | 2.859 | 0.904 |
| bio2token | 1.069 | 0.963 | 0.998 | 0.966 | 1.367 | 0.960 | **0.987** | 0.958 | 1.201 | 0.949 |
| Kanzi (30M) | 0.996 | 0.973 | 0.889 | 0.981 | 1.123 | 0.980 | 1.074 | **0.972** | **1.165** | **0.969** |

Table 2: Full backbone reconstruction. While the gap is less pronounced than $C\alpha$ only, Kanzi consistently achieves the best or second-best reconstruction across datasets. rFPSD is a $C\alpha$ only metric by construction, so it is excluded.

biased towards the capabilities of current folding models, and recent work has shown that RMSD is not a strong predictor of generative capabilities. rFPSD captures the statistics of the distribution, and our results in Table 1 suggest it may be a useful addition to the metrics used for tokenizer development. rFPSD extends the original FPSD metric introduced for measuring generative capabilities in (Geffner et al., 2025).

**[ss]RMSD** considers RMSD only over proteins with $> 60\%$ of a particular secondary structure (ss=$\alpha$, $\beta$, $c$) content. This isolates the failure modes of tokenizers and can help measure a tokenizer's ability to model unstructured regions, which is particularly important for many downstream therapeutic tasks.

We consider auxiliary metrics only over the CATH dataset, for two reasons. We do not want to use the AFDB, as we do not want to further bias metrics towards distributions over synthetic structures. Second, most other natural protein structure datasets are not large enough to provide accurate estimators for distribution metrics such as rFPSD.

We train both $C\alpha$ only and full backbone tokenizers, as both are useful depending on the task at hand. These results are shown in Tables 1 and 2. Kanzi consistently performs best or second best across dataset categories, despite being trained entirely on synthetic data. The gap is more modest for full-backbone tokenization, but we again note the significant difference in model and data scale between Kanzi and structure tokenizers like ESM3 and DPLM2 (see Figure 4). The gap in rFPSD between ESM3 and DPLM2 underscores the value of distribution-level metrics for assessing and improving tokenizer performance: while DPLM2 underperforms ESM3 on reconstruction, it surpasses it on rFPSD. As prior work has emphasized that strong reconstruction does not necessarily imply high-quality generation (Geffner et al., 2025; Hsieh et al., 2025), developing better metrics to capture generative ability during tokenizer training is an important contribution.

## 4.2 GENERATION

The primary purpose of tokenization is for downstream representations. To evaluate Kanzi's performance on structure generation, we train an autoregressive model on tokenized sequences from Kanzi. To broaden the scope of our comparisons, we also train additional autoregressive models on the DPLM and ESM tokenizers. The full details on these models is in Appendix I.2. These comparisons ensure that strong results from our tokenizer can be decoupled from the choice of generative model (e.g., autoregressive vs discrete diffusion). The full results for ESM3, DPLM-2, ESM3-AR, and DPLM2-AR are presented in Table 3. Despite having a significantly smaller parameter count than alternative models, Kanzi-AR exhibits strong performance across quality and diversity metrics.

To demonstrate a unique value of the non-invariant approach, we train a generative model on Kanzi tokens conditioned on a cryoET volume map. This is an instance where other tokenizers immediately collapse, as the conditioning signal is intrinsically non-invariant. Appendix D describes the task, demonstrates our approach, and provides quantative evidence of the utility of Kanzi tokens.

Kanzi-AR overpredicts alpha helices, a known issue for models that heavily rely on synthetic data. This can be resolved by additional post-training steps (as done in Huguet et al. (2024a)), which we leave to future work. We also noticed that evaluations across models have a substantial amount of variance depending on sampling; for instance, our benchmarking on DPLM-2 outperforms the scRMSD results reported in Wang et al. (2024) but underperforms the reported scTM values. To ensure reproducibility, we describe our generation and benchmarking process for all models in Appendix I.2.

| Model | QUALITY | | | DIVERSITY | | | | |
|---|---|---|---|---|---|---|---|---|
| | Designability ($\uparrow$) | scRMSD ($\downarrow$) | scTM ($\uparrow$) | Diversity ($\downarrow$) | Novelty ($\downarrow$) | $\alpha\%$ | $\beta\%$ | $c\%$ |
| ESM3 (650M/32) | 0.476 | 10.898 | 0.702 | 0.705 | 0.730 | 82.9 | 5.1 | 11.9 |
| ESM3 (650M/256) | 0.460 | 6.959 | 0.7199 | 0.604 | 0.692 | 74.9 | 11.5 | 13.6 |
| ESM3-AR (300M) | 0.520 | 4.252 | 0.804 | **0.241** | 0.751 | 38.6 | 16.8 | 44.6 |
| DPLM2 (650M) | 0.486 | **3.314** | **0.814** | 0.263 | 0.735 | 42.5 | 15.2 | 42.3 |
| DPLM2-AR (300M) | 0.320 | 8.989 | 0.706 | 0.308 | 0.772 | 41.2 | 18.0 | 40.8 |
| **Kanzi-AR (250M)** | | | | | | | | |
| ($\eta$=0) | 0.328 | 4.210 | 0.724 | 0.271 | 0.715 | 71.9 | 6.5 | 21.6 |
| ($\eta$=0.66) | 0.562 | 3.781 | 0.795 | 0.408 | 0.773 | 88.7 | 0.7 | 10.7 |
| ($\eta$=0.66, BoN) | **0.617** | 3.655 | 0.807 | 0.386 | 0.763 | 88.2 | 0.8 | 11.0 |

Table 3: Generative evaluation across models. **Bold** = best (per column), underline = second-best. The first ESM3 records are with 32 and 256 steps, respectively, (i.e., for the latter, every pass decodes a single new token). Kanzi-AR consistently shows strong performance across metrics. As an additional contribution, we use best-of-N sampling ($N = 2$) with log-likelihoods as our reward proxy to improve performance, demonstrating that our autoregressive prior learns a meaningful distribution.

## 4.3 ABLATIONS AND DESIGN CHOICES

This section contains a curated list of ablations and design choices we made. We present a more thorough list of findings in the appendix. Each bolded item has a corresponding reference in Figure 5.

**Encoders need token mixing for generation, but not for reconstruction.** When operating on coordinates without any invariant transforms, because the input itself carries raw positional information, it isn't obvious that pooling of local information should occur in the same way as it does in image tokenizers or invariant protein structure tokenizers (see Section K.2 for more discussion and Ellmen et al. (2025) for a deeper investigation). We ablate the window size on encoder transformers – full attention, window size 8, window size 4, and window size 0 (corresponding to a point-wise MLP on the raw coordinates). Surprisingly, the latter suffices for good *reconstructions*, but substantially harms downstream generations.

**Codebook utilization is emergent.** A surprising but reproducible phenomenon was the emergence of high codebook utilization after extended training. At the start of training, the raw coordinates are highly correlated, which in FSQ leads to low usage. Over time, this spreads out, signifying

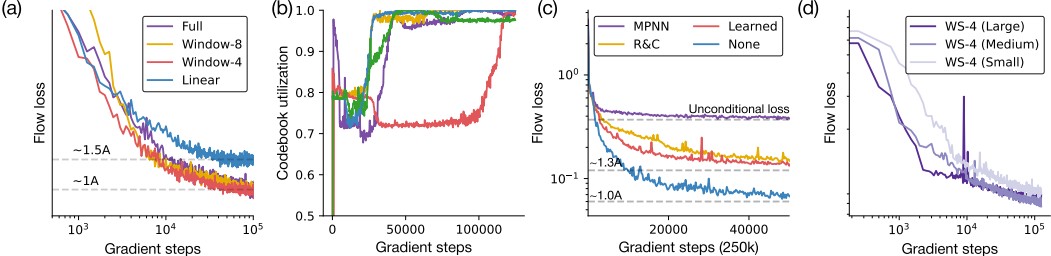

Figure 5: All flow losses use the same noise schedule; as numerical values are uninterpretable, we omit them. (a) **Encoders need mixing:** A point-wise encoder can achieve remarkably strong reconstructions (though underperforms token mixing). Downstream generative performance, however, is poor (see Appendix). (b) **Codebook utilization is emergent**. Across a large number of architectural classes, codebook usage shoots up after a large number of gradient steps. (c) **Invariant representations struggle.** Common encoders like MPNNs lead to codebook collapse and identical performance as an unconditional model. While other invariant encoders work, they underperform simply learning the pose. (d) **Flow autoencoders are scalable.** Larger model sizes converge to the same or lower loss in fewer gradient steps. Plot is log-log.

an increase in codebook utilization (see Appendix G for additional approaches we tried to increase codebook usage).

**Invariant representations struggle.** Most existing tokenizers use an invariant encoder; we experiment with two variants of this design choice – that is, we use an invariant encoder to inform an *equivariant* decoder. We experiment with a learned relative rotation between the encoder and decoder, which encourages the model to learn an invariant representation, and an explicitly invariant input. Both cases underperform simply allowing the model to tokenize the pose for both reconstruction and generation. Most graph-based invariant models (like MPNN, a common choice for proteins) lead to codebook collapse and the same flow loss as a purely unconditional model.

**Flow autoencoders are scalable tokenizers.** An advantage of transformer layers that use standard attention is their scalability. While this is computationally challenging to explore, small-scale experiments we performed ($<$0.2B parameters) consistently showed that larger models reach the same performance as smaller models after fewer steps, and model performance continues to improve with extended training. We observed no evidence of overfitting, and continued to see performance improvements well over 100k gradient steps during training.

## 5 OUTLOOK AND LIMITATIONS

In this work, we present a new approach for structure tokenization. We demonstrate that for the task of protein structure tokenization, flow autoencoders that utilize standard attention simplify model training while enjoying performance equal to or better than other state-of-the-art generative models. An autoregressive model trained on Kanzi tokens is, to our knowledge, the first tokenized structure model competitive with structure tokenizers like ESM3 or DPLM2 that use large-scale pre-training. Additional work is necessary to close the performance gap between models that leverage tokenization for structure generation, as described here, and state-of-the-art diffusion models.

Our work has several key limitations. We primarily train on the AFDB, which is synthetically biased; our introduction of the rFPSD metric was an attempt to measure these effects. For computational reasons, our models are quite small. It is challenging to know how these models will actually perform as data size, model size, and training time are increased, as was studied in Geffner et al. (2025). Similarly, we only train on proteins of size $< 256$. Geffner et al. (2025) showed that a fine-tuning stage with larger proteins could dramatically improve long protein designability. We expect that with appropriate computational resources, our models might realize similar gains. Our generative models are trained on $C\alpha$ tokenizers, a choice driven by computational limitations. During training, we were surprised to observe only small differences between $C\alpha$ and full backbone tokenizers. Extending generation to the full-backbone and all-atom case is an important future direction.

We close by noting that while diffusion/flow matching models remain state-of-the-art for protein generation, tokenization has much to offer structural biology and protein design. Biology is filled with diverse representations – cryoEM and cryoET images, single cell data, multiplexed immunofluorescence data, structural and natural language descriptions of proteins, etc. Tokenized representations are uniquely amenable to multimodal tasks beyond generation. We believe that much like the interplay between images, text, video, and audio, building large foundation models for the life sciences will require robust tokenized representations across data modalities.

ACKNOWLEDGMENTS

R.D. thanks Samuel Arnesen, Markus Marks, Laura Luebbert, Sinan Ozbay, William Arnesen, Nicholas Ritter, and Gohta Aihara for technical support and helpful feedback on the draft. R.D. thanks Atlantic Labs for insightful conversations. R.D. thanks Jeff Song and Zhen Chen for discussions on the cryoET methods described in this paper. We thank Modal for supporting this work via a Modal compute grant. This work was supported by an in-kind Landing Fellow Ship.

## 6 REPRODUCIBILITY STATEMENT

We have taken several steps to ensure reproducibility of our results. Our training relies entirely on public data (Varadi et al., 2022). The main text describes the core model architecture (Section 3.1) and training objective. We fully describe all hyperparameters in Appendix F.1 and dataset processing steps in Appendix F.2. We describe our evaluation metrics with code references in Appendix C.1 and Appendix C.2. Source code and instructions for reproducing all experiments will be released publicly after the review period, once anonymization and cleanup are complete.

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

## A  LLM Usage Statement

Large Language Models were used for polishing grammar, finding typos, and creating plotting code.

## B    ETHICS STATEMENT

AI for biology has the potential to significantly improve human health, but can also be used for ill. We strongly support the principles outlined in Responsible AI x Biodesign.

## C    METRICS

### C.1    STANDARD METRICS

This section describes all of the metrics we use in assessing model performance throughout the paper. To encourage reproducibility, we release all of these along with any relevant data with our code submission.

**RMSD:** We use the Kabsch algorithm, for which there are numerous standard implementations, to align two structures before computing RMSD.

**TM-score:** We use `biotite`'s implementation in `biotite.structure.tm_score` to compute the TM score between predicted and sampled structure. There are small differences between the biotite implementation and the `tmtools` implementation, but in our experiments these were on the order of $\approx 0.01$, so we used the former as it parallelizes more easily.

**Designability:** We follow standard practice and use ProteinMPNN in $C\alpha$ only mode to inverse fold eight putative sequences, then use ESMFold to fold those into eight output structures. We compute the RMSD/TM score between all eight and report the lowest/highest one.

**Diversity:** We compute the TM-score between all pairs of *designable* structures. A challenge with autoregressive models is they produce variable sequence lengths, which can bias diversity results to lower outputs. To mitigate this and provide a fair comparison without needing to generate thousands of structures, we take all pairs within ten residues in size. We report the average over all pairwise comparisons.

**Novelty:** We report the average TM score of the closest match in the CATH reference database over all designable structures. We found searching over the full PDB to be computationally too expensive on our hardware.

### C.2    AUXILIARY METRICS

**[ss]RMSD:** Alpha helices are highly structured and quite easy for models to reconstruct, beta sheets less so, and coils the most unstructured secondary structure element. This metric computes RMSD on proteins that are more than $60\%$ one type of secondary structure to provide a more fine-grained view of reconstruction capabilities.

**rFPSD:** We compute rFPSD over the CATH dataset by taking features from the trained GearNet fold classifier from Geffner et al. (2025). We have two datasets of features, one from the original CATH dataset and one from the tokenized and reconstructed CATH dataset. We fit Gaussians to these features using standard estimators and compute the Wasserstein distance as

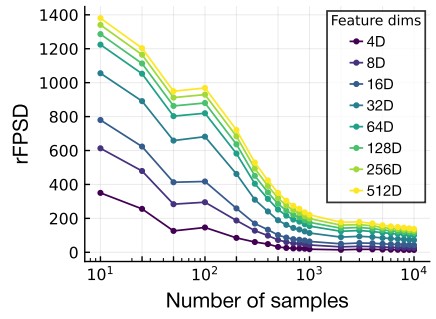

Figure 6: rFPSD converges after 5k samples.

$$\text{rFPSD} = \|\mu_r - \mu_c\|_2^2 + \text{Tr}\Big(\Sigma_r + \Sigma_c - 2(\Sigma_r \Sigma_c)^{1/2}\Big)$$

where the subscripts $c$ and $r$ reference the fit CATH and reconstructions respectively.

FPSD was originally introduced in Geffner et al. (2025). A similar metric was introduced in Faltings et al. (2025). We choose the former over the latter primarily as the latter uses the ESM3 structure

tokenizer, which is trained on a large number of synthetic structures and carries all of the biases of tokenization. While not conclusive, the observation that DPLM-2 has a lower rFPSD than ESM3 (but a higher RMSD) is encouraging for the notion that we can hill-climb rFPSD to develop tokenizers with better generative capabilities.

**Why do we only measure rFPSD on CATH?** The goal of our additional metrics are to provide measurements of quality beyond per-sample quality, which RMSD and TM-score provide, and to correct for biases endowed by model and data choices. The estimators for the Gaussian fitting are

$$
\mu = \frac{1}{N} \sum_{i=1}^{N} x_i \qquad \Sigma = \frac{1}{N-1} \sum_{i-1}^{N} (x_i - \mu)(x_i - \mu)^{\mathsf{T}}
$$

For Gaussian noise, both estimators are unbiased and obey the Central Limit Theorem. We empirically estimated that we required a minimum of 5,000 samples to get low variance estimators. We show this in Figure 6. The CATH dataset provides both the diversity and sample count required. We explicitly did not want to use synthetic data to avoid biases like overprediction of alpha helices.

## D  CRYOET RECONSTRUCTION WITH KANZI TOKENS

This section describes a conditional generation task. Cryo-electron tomography (cryoET) is a rapidly advancing imaging technique that creates 3D models of frozen biological samples in a cell (as opposed to purified protein samples). This allows biologists to understand how proteins and other biological complexes function in their natural state.

cryoET data analysis follows two stages. In the first stage, noisy 2D images are compiled to a 3D volume. A number of approaches in the computational imaging literature have approached this problem as a reconstruction task. In the second stage, a user must determine the actual proteins corresponding to a cleaned 3D sample. The input is an $L \times L \times L$ voxel mask, and the desired output is a sequence of protein structures $L \times 3$. Crucially, no sequence information is known, and there may be multiple proteins present (each of unknown length). The standard approach involves searching through extensive proteomic databases and docking proteins using the cross correlation scores to a simulated molmap. This is computationally inefficient and quite brittle.

We propose instead treating this as a generative problem, where the objective is to autoregressively generate a protein structure conditioned on the voxel cryoET data. A critical insight is that SE(3) invariant tokenizers fail to use the conditioning signal (a form of posterior collapse), since they must simultaneously reason about the generated protein and the pose indicated by the input sequence.

Figure 7 schematically depicts both our approach and performance at various protein lengths on a synthetic dataset. For each protein, we simulate a molmap using standard tools, which we then encode (using either a 3D Unet or a surface encoder). This sequence of tokens is provided as in-context conditioning to a language model trained using Kanzi tokens. Even at longer protein lengths, the resulting generated structures have $< 3\text{Å}$ RMSD agreement with the ground truth protein structure, well within the best docking approaches.

## E  REPRESENTATION QUALITY

Table 4 evaluates Kanzi tokens on a subset of representation quality benchmarks from Yuan et al. (2025). The subset of tasks we consider are residue level tasks designed to probe the information capacity of individual tokens. We defer a full description of the tasks to Yuan et al. (2025); in summary, we include binary classification tasks for binding, catalytic, conserved, epitope, and repeated residues, and regression tasks for measuring flexibility of regions. For each task, we take post-quantized representations and use a 2-layer probing MLP. FlexRMSF probing additionally includes a sigmoid layer mapping outputs from 0 to 1.

While Kanzi tokens show some representative capabilities, both Kanzi and bio2token underperform invariant models. This is expected, since most residue probing tasks are fundamentally local tasks. The probing layer must learn to convert global residue information to a local estimator, which is challenging for a simple two-layer MLP.

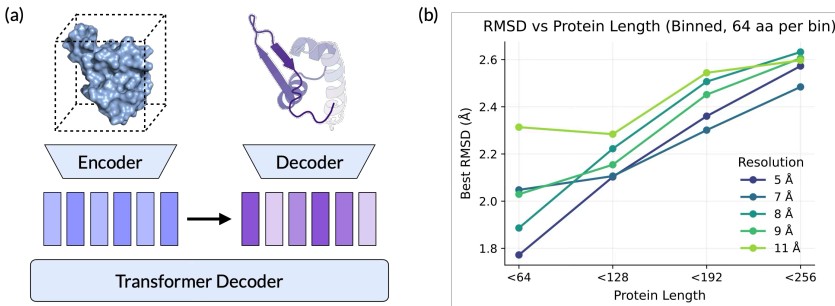

Figure 7: (a) Schematic of our approach to cryoET docking. A volumetric map is encoded to a sequence of tokens, which are provided as in-context conditioning to a language model trained to output Kanzitokens. These tokens are decoded to the original protein. (b) Across simulated resolutions and protein lengths, this approach consistently achieves $3\text{\AA}$ reconstruction.

| | bindint | | catint | | con | | ept | | rep | | phys | |
|---|---|---|---|---|---|---|---|---|---|---|---|---|
| Model | Fold | Family | Fold | Family | Fold | Family | Fold | Family | Fold | Family | Fold | Family |
| bio2token | 0.489 | 0.637 | 0.544 | 0.539 | 0.512 | 0.543 | 0.514 | 0.525 | 0.503 | 0.569 | 0.312 | 0.268 |
| dplm2 | 0.540 | 0.794 | 0.598 | 0.703 | 0.570 | 0.728 | 0.623 | 0.700 | 0.507 | 0.763 | 0.475 | 0.433 |
| esm3 | 0.497 | 0.787 | 0.549 | 0.749 | 0.541 | 0.647 | 0.614 | 0.644 | 0.519 | 0.677 | 0.433 | 0.414 |
| kanzi | 0.504 | 0.692 | 0.531 | 0.603 | 0.525 | 0.626 | 0.532 | 0.604 | 0.526 | 0.614 | 0.298 | 0.357 |

Table 4: Updated performance table with revised Kanzi benchmarking. AUROC for all tasks except *phys*, which uses Spearman's $\rho$.

There are several important caveats about these experiments. First, we were unable to directly reproduce the results in Yuan et al. (2025) using the publicly available checkpoints and codebase. We use the processed datasets provided and train probing layers for 500 epochs with the AdamW optimizer, learning rate $10^{-4}$. Despite this, there are still differences in our results for ESM3 compared to Yuan et al. (2025). We will make training/benchmarking code available upon de-anonymization.

# F  MODEL

This section contains details on the full model training pipeline.

## F.1  HYPERPARAMETERS

We train with AdamW, $\beta_1 = 0.9, \beta_2 = 0.95$. We train tokenizer models with the following configurations (bold indicates base models used for most experiments). We train most models on 1-2 H100s (when training with two GPUs, we use 4 micro-steps for an effective batch size of 256). For each micro step, the elements in the batch are composed of a single protein with randomly augmented views. This allows us to avoid masking/batching with different sequence lengths.

## F.2  DATA PREPROCESSING

We train on the Foldseek-clustered AFDB $\mathcal{D}_{\text{FS}}$. We perform the following filtering steps

1. We remove all chains with coil percentage $> 70\%$.

2. We remove all chains with mean residue-wise pLDDT $< 80$.

3. We keep only chains where 80% of the residues have pLDDT $> 70$.

4. We filter all structural homologs of chains in our test sets: a subset of the AFDB (already filtered), CAMEO, CASP14, CASP15, and a subset of CATH.

| Parameter | Value |
|---|---|
| Batch size | 32 |
| Encoder layers | 2 |
| Decoder layers | **8**/12 |
| Attention heads | 8 |
| Encoder channels | **256**/384/512 |
| Decoder channels | 256/**512**/784 |
| Pair-bias channels | **64** |
| FSQ levels | (8, 5, 5, 5) |
| MLP factor | 4 |
| Dropout | 0.1 |
| Learning rate | $1.7 \times 10^{-4}$ |
| Warmup iters | 1000 |
| LR decay iters | 100000 |
| Minimum LR | $1 \times 10^{-4}$ |
| Gradient clip | 1.0/None |
| Sliding window size | 4/**8**/16/None |
| Micro-steps (grad accum.) | 8 |
| QKNorm | True/**False** |

Table 5: Training configuration hyperparameters.

The first two requirements are fairly important to ensure high quality structures. We did not ablate the effect of the last step; various other works like Bose et al. (2023) have imposed similar filtering criteria.

During training, we mean-center all coordinates using $C\alpha$ position (we do this for full backbone data as well).

It is easy to see that one cannot have a diffusion process that is invariant to translations, since the probability mass would not integrate to one (Garcia Satorras et al., 2021). It is standard to instead define a diffusion process over a *subspace* of $\mathbb{R}^n$, i.e., the zero center-of-mass subspace. The noise interpolation in flow matching is thus well defined, since if $\epsilon \sim \mathcal{N}(0, 1)$ and $\mathbf{x}$ both have center-of-mass zero, then $\mathbf{x}_t = t\mathbf{x} + (1 - t)\epsilon$ also does by linearity (Hoogeboom et al., 2022).

We augment proteins with random rotations during training, computed using the following snippet:

```
from scipy.spatial.transform import Rotation
def sample_uniform_rotation(shape=tuple()):
    return torch.tensor(
        Rotation.random(prod(shape)).as_matrix()
    ).reshape(*shape, 3, 3)
```

### F.3 INFERENCE

**Diffusion sampling:** As mentioned in the main text, we use classifier-free guidance, score annealing, and noise annealing. For the most part, we take the parameters from Proteina without further exploration. A major advantage of the flow autoencoder paradigm is the noise/score scale can be tuned to the particular task at hand. **Best-of-N sampling:** In Table 3, we include a single result using best-of-2 sampling. Best-of-N sampling is a unique strength of autoregressive models; in each forward pass, we store the log-likelihood, and decode the sequence with the best log-likelihood. The fact that we have a computationally inexpensive closed-form log-likelihood is a unique strength of autoregressive models; computing log-likelihoods requires a full ODE integration for flow models. In other words, the log-likelihood gives us a free estimator for the downstream model quality before running the diffusion process or designability checks. We include this primarily to demonstrate that our learned discrete codebook encodes useful knowledge about the token quality.

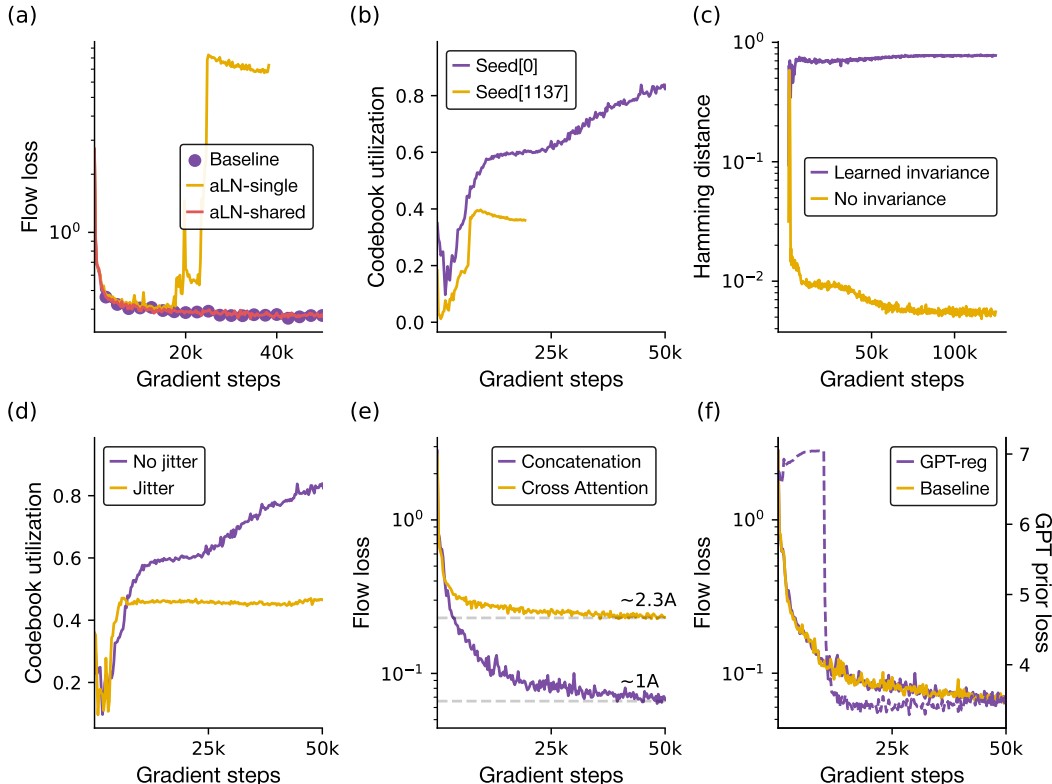

Figure 8: (a) adaLN-single trains unstably, but sharing the weights across all layers matches baseline performance and reduces parameter count by 30%. (b) Two runs at different starting seeds with learned augmentations show very different codebook utilizations. (c) Rotational augmentation increases the Hamming similarity between different views of the same protein, which by default are totally different sequences. (d) Codebook jitter mostly just reduces codebook utilization. (e) Cross attention unambiguously makes performance worse. (f) The baseline and GPT-regularized version have identical flow losses, but the GPT-regularized version has an additional per-token cross-entropy loss (the baseline version stays at the random initialization $\log(1000)$).

A valid concern with best-of-N sampling is mode collapse; the strong diversity metrics in Table 3 emphasize that this is not the case with our results.

## G  ABLATIONS

This section contains a list of all ablations that we conducted in addition to those presented in the main text. We highlight ablations with **positive** results in purple, **negative** results in red, and **ambiguous** results in blue. [1]

**AdaLN:** Standard diffusion models use adaptive layer normalization to condition the model on time, but this time conditioning relies on thick linear layers and takes 27% of all model parameters. Subsequent work (Chen et al., 2023) used a variant, adaLN-single, which uses a single adaLN MLP

---

[1]Warning: For execution speed, we often relied on the flow loss as proxy for downstream performance. One has to be careful while doing this. The comparison is valid and correlates well with reconstruction for *identical* noise schedules and invariance/equivariance setups, but this is no longer true when encoder symmetry properties change. An encoder that does not learn the pose will almost always have a higher flow loss than one that does learn the pose, even if the former has better reconstructions. In all of the ambiguous cases, we check reconstructions on about 1k CATH structures. To aid the interested reader, we present flow losses in the plots associated with each ablation and annotate approximate reconstructions. We often omit the actual numerical values of the flow loss, since it is effectively uninterpretable as a numerical value (though extremely useful to compare performance across experiments).

on the first layer and has lightweight projections for all subsequent layers. We explore a third option, where a single MLP is shared across *all* layers. We find that while adaLN-single trains very unstably, weight sharing across layers provides equal performance as the standard DiT approach. **We add adaLN weight-sharing to Kanzi**.

**Learned rotational invariance:** To learn rotational invariance, we add an additional random relative rotation between the encoded protein and the encoded noise. This encourages the model to learn an invariant representation, a fact we confirm by observing the Hamming and MSE distances between the learned vectors. These models can train, but are extremely stochastic. Intuitively, the model needs to learn how the conditioning vector can relate in very arbitrary ways to the diffusion process. We ran trials at several random seeds and witnessed very different codebook utilizations. That coupled with the reconstruction performance being worse on CATH by about 0.4Å made us reject this change. **We exclude learned rotational invariance from Kanzi.**

**Cross-attention conditioning:** Some work suggests that cross attention may be more effective at conditioning diffusion transformers for more complex conditioning sequences (Chen et al., 2024). We explore conditioning our decoder with cross attention, where the input is a six-eight layer encoder. Our cross attention layers follow standard practices and attend to the main diffusion trunk following each self-attention layer (Vaswani et al., 2017; Peebles & Xie, 2023). **This unambiguously hurts performance, so we exclude it from the model.**

*Curriculum learning:* Since the non-invariant version worked quite well, we thought we could transition slowly between a non-invariant and a learned invariant version by gradually augmenting with larger and larger rotations during training (using spherical linear interpolation). This did not help at all and mostly just harmed codebook utilization and flow loss. **We exclude curriculum rotational invariance from training**.

**Explicit rotational invariance:** Similar to above, it seems reasonable that an invariant representation could condition a non-equivariant decoder. As we discuss in Section 4.3, certain invariant encoders like MPNN (Dauparas et al., 2022), a common choice for protein design tasks, immediately led to codebook collapse and poor performance on par with unconditional flow models. We explored other variants. In particular, we embedded the distances between coordinates $d_{ij}$ into feature vectors, giving inputs with shape $L \times L \times d$. We applied successive layers of row and column attention, then took a mean down a sequence axis as the conditioning input. This worked slightly better in that the model didn't immediately collapse, but didn't really provide any tangible benefit beyond the aesthetic benefit that different poses would encode to the same protein. **We exclude explicit rotational invariance from the model.**

**Low codebook utilization:** FSQ typically has very high codebook utilization (Mentzer et al., 2023), but as our data is low dimensional and highly correlated at input, many values ended up in the same "bin" which led to low usage. We explored a variety of options to increase codebook utilization. As mentioned in the main text, we eventually realized that we could get high codebook usage by just letting the model train longer, but we leave the discussion here as a potentially useful record of ablations.

*Codebook jitter:* One strategy to encourage higher codebook use is to "jitter" the codewords. This is typically used in VQVAE; for FSQ, we add a small noise term before bin quantization. This noise term was quite small but generally made results worse. **We exclude codebook jitter from model training.**

*Rotational transforms:* Recently, Fifty et al. (2024) saw improvements in tokenization from applying a generalized rotation before and after the quantization step. This seemed to add complexity but have basically no effect on our task specifically, so **we exclude rotational transforms from the model**.

*Non-linear embeddings:* We initially started with just a linear upsampling layer, as is done in Geffner et al. (2025). Since linear layers preserve correlations in highly structured ways, we went from `Linear` to `Linear -> Swish -> Linear -> LayerNorm`. We ablated all of these carefully; the addition of both the non-linearity and the layer normalization both seem important. **We add non-linear point-wise encodings to Kanzi.**

**GPT prior:** Prior works have suggested adding a small GPT-regularization term to encourage codebooks to learn tokens more suitable for autoregressive, next-token-prediction modeling (Radford et al., 2018; 2019; Vaswani et al., 2017). We ablate training a small GPT model (2 layers, dimension

| Dataset | Reconstruction ($\mathring{A}$) |
|---------|-------------------------------|
| CATH | 1.010 |
| CASP14 | 0.835 |
| CASP15 | 0.909 |
| CAMEO | 0.898 |
| AFDB | 1.051 |

Table 6: Long sequence ablation. Sliding window attention with window size 8 maintains strong reconstruction across datasets.

512) in tandem with our tokenizers. The cross-entropy GPT loss is added as a regularization term to the flow matching loss. We did not fully ablate these results on generative capabilities, but **the autoregressive regularization did not harm reconstructions.**

**Long sequences:** To test whether the strong performance of sliding window attention sustains at longer sequence lengths, we trained an identical Kanzi model on sequences up to 512 residues. We find that we maintain strong performance across all reconstruction benchmarks, see Table 6.

### G.1 PAIR-BIASING AND SELF-CONDITIONING

Pair-biasing is a technique where the attention scores $QK^T$ in a transformer are biased by projecting learned pair features $\mathbf{z}_{ij} \in \mathbb{R}^{L \times L \times d}$ in each layer. These are very common in protein design models since their introduction in AlphaFold2 (Jumper et al., 2021; Abramson et al., 2024). However, AlphaFold2 and AlphaFold3 both use triangular updates, which many subsequent works forgo due to their computational cost. Thus, it is actually quite unclear if pair-biasing is actually helpful when used *without* the additional use of triangular attention.

Similarly, self-conditioning (described in Stärk et al. (2023) and Chen et al. (2022)) conditions a flow model on previous predictions. Instead of a flow field $\mathbf{v}_\theta(\mathbf{x}_t, t, \hat{\mathbf{c}})$, we now have a flow field $\mathbf{v}_\theta(\mathbf{x}_t, \tilde{\mathbf{x}}_1, t, \hat{\mathbf{c}})$, where $\tilde{\mathbf{x}}_1$ is a coarse prediction of the clean data from the previous timestep. Self-conditioning is implemented during training by providing a coarse prediction 50% of the time; the remainder of the time, we simply drop out the prediction (i.e., we do $\mathbf{v}_\theta(\mathbf{x}_t, \varnothing, t, \hat{\mathbf{c}})$). Various works have reported improved performance by using self-conditioning.

We found it difficult to fully disentangle the effects of self-conditioning and pair biasing. A complicating challenge was we ultimately care about downstream generative performance, which reconstruction alone does not track with. For this reason, we conducted a series of ablations on pair-biasing (no triangular updates), self-conditioning, and the two combined, all for a small 11M parameter model. These results are summarized in Table 7 and visualized in Figure 10. Our primary conclusion was that while pair-biasing and self-

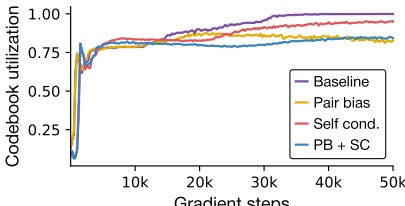

Figure 9: Codebook usage decreases with pair-bias and self-conditioning

conditioning seemed to help reduce both RMSD and TM-score, the effect was not enormous, and our 30M models without either addition were already at 1Å resolution or better. Moreover, self-conditioning (possibly due to the large 50% dropout probability) tended to harm codebook utilization. While codebook utilization is not a good per se, it is a useful measure to track. This drop is shown in Figure 9. Given these complexifiers, **we did not include self-conditioning and optionally included pair-biasing in our models**. However, we think exploring self-conditioning and pair-biasing, and how they impact generative capabilities, is a potentially fruitful future avenue of research.

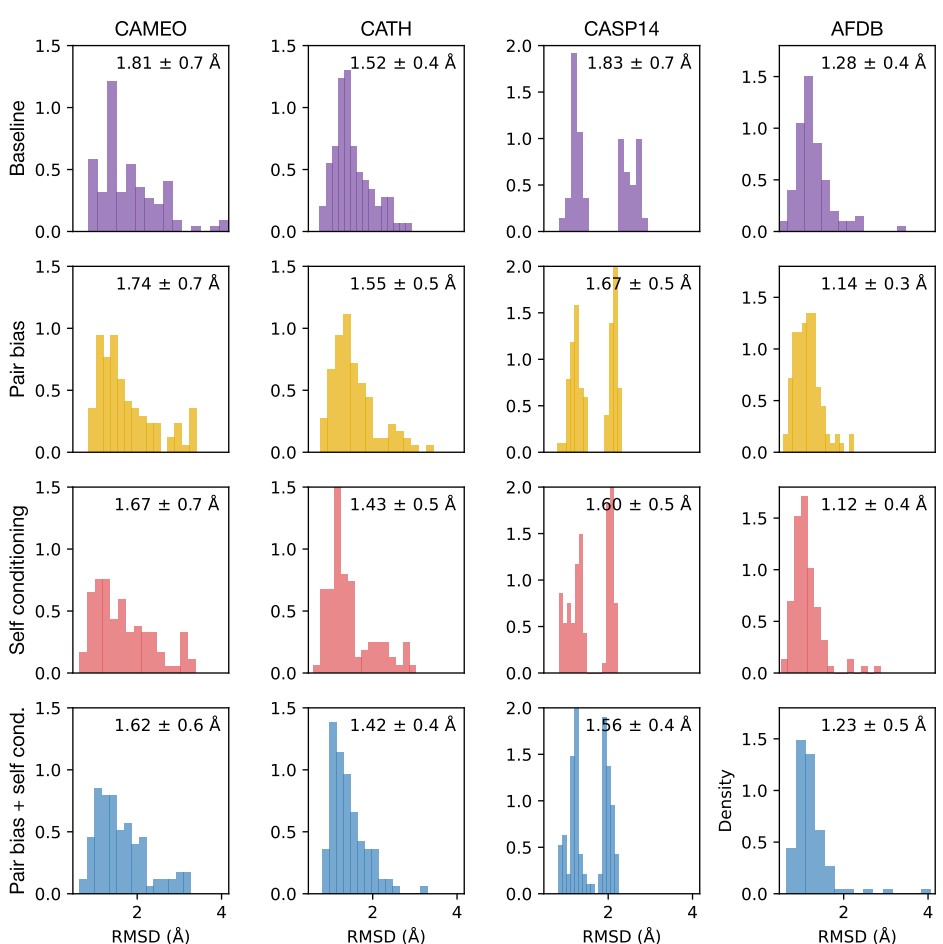

Figure 10: Histogram of RMSDs for pair bias and self-conditioning ablations across datasets.

| Method | RMSD | TM-score |
|---|---|---|
| **CAMEO** | | |
| Baseline | $1.812 \pm 0.702$ | $0.932 \pm 0.053$ |
| Pair bias | $1.741 \pm 0.671$ | $0.933 \pm 0.053$ |
| Self-cond | $1.670 \pm 0.679$ | $0.938 \pm 0.051$ |
| Pair bias + Self-cond | $1.619 \pm 0.602$ | $0.943 \pm 0.042$ |
| **CATH** | | |
| Baseline | $1.519 \pm 0.446$ | $0.953 \pm 0.027$ |
| Pair bias | $1.551 \pm 0.534$ | $0.947 \pm 0.037$ |
| Self-cond | $1.431 \pm 0.543$ | $0.957 \pm 0.030$ |
| Pair bias + Self-cond | $1.420 \pm 0.422$ | $0.958 \pm 0.028$ |
| **CASP14** | | |
| Baseline | $1.833 \pm 0.665$ | $0.948 \pm 0.032$ |
| Pair bias | $1.674 \pm 0.465$ | $0.954 \pm 0.021$ |
| Self-cond | $1.596 \pm 0.473$ | $0.960 \pm 0.018$ |
| Pair bias + Self-cond | $1.561 \pm 0.434$ | $0.964 \pm 0.016$ |
| **AFDB** | | |
| Baseline | $1.282 \pm 0.421$ | $0.964 \pm 0.023$ |
| Pair bias | $1.141 \pm 0.318$ | $0.972 \pm 0.014$ |
| Self-cond | $1.122 \pm 0.356$ | $0.972 \pm 0.017$ |
| Pair bias + Self-cond | $1.233 \pm 0.459$ | $0.965 \pm 0.028$ |

Table 7: Pair-bias and self-conditioning ablation summary statistics (mean $\pm$ std) for RMSD and TM-score across datasets.

## H  OVERVIEW OF FLOW MATCHING

Flow matching is extensively described elsewhere: see Lipman et al. (2022) or Albergo & Vanden-Eijnden (2022) for original presentations and Esser et al. (2024) for a more applied discussion. Flow matching generalizes diffusion models (Song et al., 2020; Ho et al., 2020), which similarly interpolate between two distributions. While the distinction between flow matching and diffusion is often stated to be straight vs curved paths, this is more of a property of the sampler being used (see Gao et al. (2024) for a good discussion). As mentioned earlier, for Gaussian flows on $\mathbb{R}^n$, we have closed-form expressions to convert between the score and the learned vector field. For this reason, we often use the terms interchangeably. Flow objectives are a good fit for many molecular diffusion tasks because they avoid the instabilities at early or late timesteps (images solve this problem by clipping, which is not as easy to do for points in $\mathbb{R}^3$).

For completeness, we present the standard flow matching objective and parameterization in notation consistent with the remainder of the paper. Given a data distribution $p_1$ and an easy to sample prior $p_0$, we wish to learn a family of functions $\mathbf{u}(\mathbf{x}_t, t) : \mathbb{R}^d \times [0, 1] \rightarrow \mathbb{R}^d$ that would push from the distribution $p_0$ to $p_1$. If we did have access to a closed-form expression for $\mathbf{u}(\mathbf{x}_t, t)$, then we could simply optimize

$$\mathcal{L}_{FM} = \mathbb{E}_{t \sim \mathcal{U}(0,1), \mathbf{x}_t \sim p_1} \left[ \|\mathbf{u}(\mathbf{x}_t, t) - \mathbf{v}_\theta(\mathbf{x}_t, t)\|^2 \right]$$

However, we do not have such a $\mathbf{u}(\mathbf{x}_t, t)$. The critical insight of conditional flow matching is that one can construct a conditional path

$$\mathbf{u}(\mathbf{x}_t, t) = \mathbb{E}_{\mathbf{x}_1 \sim p_1, \mathbf{x}_0 \sim p_0} \left[ \frac{\mathbf{u}(\mathbf{x}_t | \mathbf{x}_0, \mathbf{x}_1, t) p(\mathbf{x}_t | \mathbf{x}_0, \mathbf{x}_1, t)}{p(\mathbf{x}_t, t)} \right]$$

In this expression, we have *introduced* a family of conditional probability paths $p(\cdot | \cdot, \cdot, t)$ that induces conditional vector fields $\mathbf{u}(\cdot | \cdot, \cdot, t)$. As we do not know $p(\mathbf{x}, t)$, we still cannot regress against $\mathbf{u}(\mathbf{x}_t, t)$. However, if we define

$$\mathcal{L}_{CFM} = \mathbb{E}_{t \sim \mathcal{U}(0,1), \mathbf{x}_0 \sim p_0, \mathbf{x}_1 \sim p_1, \mathbf{x}_t \sim p(\mathbf{x}_t | \mathbf{x}_0, \mathbf{x}_1, t)} \left[ \|\mathbf{u}(\mathbf{x}_t | \mathbf{x}_0, \mathbf{x}_1, t) - \mathbf{v}_\theta(\mathbf{x}_t, t)\|^2 \right]$$

then by observing that $\nabla_\theta \mathcal{L}_{FM} = \nabla_\theta \mathcal{L}_{CFM}$, we note that we can optimize $\mathcal{L}_{CFM}$ to regress $\mathbf{v}_\theta(\mathbf{x}_t, t)$ towards the true vector field. In words, we construct a probability path between two distri-

butions, sample a point from each distribution, sample a time $t \in [0, 1]$, and use the aforementioned probability path to sample $\mathbf{x}_t$. As noted earlier, the standard choice is the linear probability path

$$p(\mathbf{x}_t|\mathbf{x}_0, \mathbf{x}_1, t) = \mathcal{N}(\mathbf{x}_t|t\mathbf{x}_1 + (1-t)\mathbf{x}_0, \sigma^2)$$

where we take $\sigma = 0$ in our experiments (though this is not strictly required, see Gao et al. (2024)). This gives rise to the simple vector field $\mathbf{u}(\mathbf{x}_t|\mathbf{x}_0, \mathbf{x}_1, t) = (\mathbf{x}_1 - \mathbf{x}_0)$.

# I COMPARISONS

This section contains further details on the models we benchmark against.

## I.1 PRIOR TOKENIZATION LOSSES

First, we describe the objectives prior works have optimized against and discuss the pitfalls of each.

**FAPE**. The frame-aligned point error (FAPE) loss has become the canonical SE(3)-invariant loss since AlphaFold2 (Jumper et al., 2021). Define a frame by a translation rotation pair $T_i = (R_i, \mathbf{t}_i)$. The action of a frame on a vector $\mathbf{x}_j$ is $T_i\mathbf{x}_j = R\mathbf{x}_j + \mathbf{t}_i$. Given predicted coordinates, predicted frames, true coordinates, and true frames $\mathbf{x}_j, T_i, \mathbf{x}_j^{\text{true}}, T_i^{\text{true}}$, the FAPE loss is defined by

$$FAPE(\mathbf{x}_j, T_i, \mathbf{x}_j^{\text{true}}, T_i^{\text{true}}) = \|T_i^{-1}\mathbf{x}_j - T_i^{\text{true}\,-1}\mathbf{x}_j^{\text{true}}\|_2 \tag{6}$$

Every point is put into the local reference frame defined by every amino acid. It is easy to show that this loss is invariant under rotations and translations, but not reflections as chirality is important for biomolecules.

FAPE was very impactful, but it has several drawbacks. It scales quadratically between the atom count and the frames. It requires clamping to train stably and can still be very unstable in early training, and thus often requires an associated binned cross-entropy version. Errors in bond angles can cause kinks in the optimization spectra; AlphaFold2 uses several supplemental bond angle and violation losses to help reduce these issues.

**dRMSD** An alternative to FAPE, used by Hayes et al. (2025), is dRMSD, where we regress against inter-atom distances within a ground truth and a predicted structure. Explicitly, let $d_{ij} = \|\mathbf{x}_i - \mathbf{x}_j\|_2$. Then dRMSD is given by

$$\text{dRMSD} = \|d_{ij}^{\text{pred}} - d_{ij}^{\text{true}}\|_2 \tag{7}$$

dRMSD is invariant under chirality since $\|\mathbf{x}_i - \mathbf{x}_j\|_2 = \|(-\mathbf{x}_i) - (-\mathbf{x}_j)\|_2$. ESM3 thus introduced an additional "backbone vector loss," whose primary purpose seems to be to break the chirality symmetry in proteins. Like FAPE, dRMSD is quadratic in atom count, and can struggle to optimize long-range contacts due to being dominated by errors from numerous short range interactions.

**RMSD** Some works, notably Liu et al. (2024) and Gao et al. (2025), directly optimize the RMSD. This requires performing a rigid alignment using the Kabsch algorithm. This has several issues. First, backpropagating through the Kabsch algorithm is non-trivial and can't be done at low precision, which makes efficient training challenging. Second, even at float32 precision, the Kabsch algorithm has discontinuities whenever singular values are particularly close. It also requires a determinant correction, which can invert the handedness of a rotation and introduce another discontinuity. These cause instabilities during training. Third, while these alignments work fine at small datasets, they tend to regress to the mean at larger scale, which makes them less suited for modeling more disordered structures and targets of therapeutic interest. Finally, the SVD based alignment is $O(L^3)$ in the sequence length $L$, though this may be faster in practice using iterative approximations to the SVD

Most other losses (such as the multiple losses used in FoldToken) are derivatives or binned versions of one of the three discussed here designed to help mitigate some of these issues.

## I.2 MODELS

**ESM-3** We pull the model as described at `https://github.com/evolutionaryscale/esm`. Several others have observed that ESM3, having been trained on a large amount of metagenomic data, struggles to generate designable sequences. Geffner et al. (2025) reports designability of $22\%$. We find that we can improve on this slightly by increasing the number of steps from the default $8$ on the GitHub up to a single token per step. We hypothesize this is partially because the ESM3 tokenizer is so high variance; even at high designabilities, the mean scRMSD remains high due to a few decoded structures with incredibly high RMSDs ($>20$ Å).

**ESM-AR** ESM-AR was trained from scratch on a mix of AFDB data and PDB data. It has dimension 1280, MLP dilation factor 2, 16 layers, and 16 heads. Otherwise, it is a standard transformer with pre-LayerNorms and GELU activation functions. ESM-AR seems to improve ESM3's designability performance, but does not continually scale with techniques like best-of-N sampling, which we again attribute to the aforementioned high variance decoding in the ESM3 decoder.

**DPLM2** We pull the DPLM2 weights/code from `https://github.com/bytedance/dplm`, and otherwise follow the generation example exactly using `generate_dplm2.py`. A potential issue we noted was while our generations had similar scRMSDs as those reported in Wang et al. (2024), we slightly underperformed the reported scTM values. The difference of $\approx 0.14$ seems large, but given that scRMSD is generally acknowledged to be a more accurate reflection of protein structure that TM, seems within reason.

The DPLM-2 structure tokenizer is fine-tuned on 220k structures sourced from the PDB + SwissProt. However, the structure encoder is a large pretrained GVP-Transformer (Hsu et al., 2022) trained on 12 million AFDB structures. We thus include these in the DPLM-2 data count for Figure 4.

**DPLM-AR** The DPLM-2 codebook has size 8192, compared to the 4096 token ESM codebook. Other than the projection and embedding layers, the DPLM-AR model is identical to the ESM-AR model. Unlike the ESM-AR model, the DPLM-AR model slightly underperforms DPLM2.

**IST** We used the 1.7k codebook tokenizer, but were unable to generate any designable structures using the generative models released with the InstaDeep Structure Tokenizer. Otherwise, we exactly followed the encoding/decoding process described in `https://github.com/InstaDeepai/protein-structure-tokenizer`.

**bio2token** We pulled weights and scripts from `https://github.com/flagshippioneering/bio2token`. As bio2token operates over point clouds, rather than frames, we tokenized and reconstructed $C\alpha$ and full backbone coordinates separately. We excluded bio2token from CATH comparisons as it is explicitly trained on the entirety of CATH.

**FoldToken4** We pull FoldToken from `https://github.com/A4Bio/FoldToken_open` and follow the described reconstruction process. The language model FoldGPT described in the original paper does not have publicly available weights.

**Other tokenizers** There are three additional tokenizers we do not explicitly benchmark against, described in Zhang et al. (2024), Zhang et al. (2024), and Lin et al. (2023). The former has significant GPU and RAM requirements that made it infeasible to do thorough benchmarks on A100s. None of the three have public models with generative capabilities, so we opt to focus on tokenizers that demonstrate downstream performance along generative axes and provide good coverage of different tokenization techniques.

## J EXTENDED RESULTS

This section contains the tables from the main text with additional error bars. Our primary observation is that transformer based tokenizers seem to have a number of catastrophic errors that lead to very high variances in the output. We will make all these experiments public along with our benchmarking repo upon acceptance and deanonymization.

| | CAMEO | | CASP14 | | CASP15 | | CATH | | AFDB | |
|---|---|---|---|---|---|---|---|---|---|---|
| | RMSD ($\downarrow$) | TM ($\uparrow$) | RMSD | TM | RMSD | TM | RMSD | TM | RMSD | TM |
| DPLM2 | $1.651 \pm 1.39$ | $0.876 \pm 0.14$ | $1.008 \pm 0.39$ | $0.951 \pm 0.02$ | $2.160 \pm 1.75$ | $0.866 \pm 0.12$ | $1.641 \pm 1.27$ | $0.897 \pm 0.09$ | $4.676 \pm 6.04$ | $0.810 \pm 0.15$ |
| ESM3 | $0.860 \pm 1.60$ | $0.955 \pm 0.09$ | $\mathbf{0.462 \pm 0.33}$ | $\mathbf{0.987 \pm 0.02}$ | $\mathbf{1.021 \pm 1.89}$ | $\mathbf{0.969 \pm 0.05}$ | $1.048 \pm 1.70$ | $\mathbf{0.957 \pm 0.07}$ | $2.384 \pm 4.08$ | $0.915 \pm 0.11$ |
| FoldToken | $2.539 \pm 3.03$ | $0.881 \pm 0.12$ | $2.194 \pm 2.90$ | $0.936 \pm 0.08$ | $6.629 \pm 8.63$ | $0.744 \pm 0.29$ | $1.298 \pm 1.57$ | $0.920 \pm 0.06$ | $2.161 \pm 1.44$ | $0.858 \pm 0.09$ |
| IST | $1.637 \pm 1.86$ | $0.916 \pm 0.13$ | $0.900 \pm 0.21$ | $0.960 \pm 0.01$ | $1.252 \pm 0.29$ | $0.953 \pm 0.02$ | $1.201 \pm 0.72$ | $0.940 \pm 0.04$ | $2.872 \pm 3.46$ | $0.862 \pm 0.11$ |
| bio2token | $1.076 \pm 0.27$ | $0.948 \pm 0.06$ | $1.006 \pm 0.24$ | $0.952 \pm 0.02$ | $1.377 \pm 0.41$ | $0.939 \pm 0.06$ | $\underline{0.993 \pm 0.25}$ | $0.942 \pm 0.04$ | $1.212 \pm 0.49$ | $0.932 \pm 0.04$ |
| Kanzi (30M) | $0.936 \pm 0.27$ | $0.948 \pm 0.04$ | $0.861 \pm 0.13$ | $0.958 \pm 0.01$ | $1.345 \pm 0.46$ | $0.951 \pm 0.02$ | $1.098 \pm 0.56$ | $0.940 \pm 0.04$ | $1.069 \pm 0.49$ | $0.947 \pm 0.03$ |
| Kanzi (30M)* | $\mathbf{0.817 \pm 0.29}$ | $\mathbf{0.960 \pm 0.03}$ | $\underline{0.698 \pm 0.14}$ | $\underline{0.972 \pm 0.01}$ | $1.267 \pm 0.55$ | $0.963 \pm 0.01$ | $\mathbf{0.953 \pm 0.57}$ | $\underline{0.955 \pm 0.04}$ | $\mathbf{0.870 \pm 0.39}$ | $\mathbf{0.962 \pm 0.02}$ |
| Kanzi (11M) | $1.016 \pm 0.31$ | $0.937 \pm 0.05$ | $0.912 \pm 0.13$ | $0.954 \pm 0.01$ | $1.259 \pm 0.33$ | $0.955 \pm 0.01$ | $1.156 \pm 0.69$ | $0.934 \pm 0.04$ | $1.210 \pm 0.69$ | $0.934 \pm 0.04$ |
| Kanzi (11M) | $0.863 \pm 0.25$ | $0.952 \pm 0.04$ | $0.762 \pm 0.10$ | $0.968 \pm 0.01$ | $\underline{1.105 \pm 0.35}$ | $\underline{0.965 \pm 0.01}$ | $0.994 \pm 0.65$ | $0.950 \pm 0.04$ | $\underline{0.994 \pm 0.46}$ | $\underline{0.952 \pm 0.03}$ |

Table 8: Reconstruction metrics across tokenizers for $C\alpha$ reconstruction with errors. Smaller datasets unsurprisingly have very large error bars. An advantage of flow tokenizers is they tend to have lower variances across the board. Other tokenizers seem to have a small number of coordinates with catastrophically high RMSDs, which leads to really large standard deviations.

| | CAMEO | | CASP14 | | CASP15 | | CATH | | AFDB | |
|---|---|---|---|---|---|---|---|---|---|---|
| | RMSD ($\downarrow$) | TM ($\uparrow$) | RMSD | TM | RMSD | TM | RMSD | TM | RMSD | TM |
| DPLM2 | $1.631 \pm 1.39$ | $0.928 \pm 0.09$ | $0.995 \pm 0.39$ | $0.959 \pm 0.04$ | $2.144 \pm 1.75$ | $0.953 \pm 0.08$ | $1.717 \pm 1.85$ | $0.925 \pm 0.08$ | $4.646 \pm 6.03$ | $0.880 \pm 0.14$ |
| ESM3 | $\mathbf{0.861 \pm 1.60}$ | $\mathbf{0.980 \pm 0.05}$ | $\mathbf{0.463 \pm 0.33}$ | $\mathbf{0.994 \pm 0.00}$ | $\mathbf{1.018 \pm 1.87}$ | $\mathbf{0.983 \pm 0.06}$ | $1.151 \pm 2.20$ | $\underline{0.971 \pm 0.06}$ | $2.378 \pm 4.08$ | $0.944 \pm 0.11$ |
| FoldToken | $2.498 \pm 3.06$ | $0.922 \pm 0.09$ | $2.323 \pm 3.01$ | $0.958 \pm 0.07$ | $6.580 \pm 8.64$ | $0.809 \pm 0.22$ | $1.352 \pm 1.84$ | $0.944 \pm 0.05$ | $2.120 \pm 1.44$ | $0.907 \pm 0.08$ |
| IST | $1.626 \pm 1.84$ | $0.954 \pm 0.08$ | $0.896 \pm 0.21$ | $0.974 \pm 0.01$ | $1.244 \pm 0.29$ | $0.975 \pm 0.01$ | $1.195 \pm 0.71$ | $0.956 \pm 0.04$ | $2.859 \pm 3.45$ | $0.904 \pm 0.11$ |
| bio2token | $1.069 \pm 0.28$ | $0.963 \pm 0.04$ | $0.998 \pm 0.24$ | $0.966 \pm 0.01$ | $1.367 \pm 0.42$ | $0.960 \pm 0.04$ | $\mathbf{0.987 \pm 0.25}$ | $0.958 \pm 0.03$ | $1.201 \pm 0.49$ | $0.949 \pm 0.03$ |
| Kanzi (30M)* | $\underline{0.996 \pm 0.31}$ | $\underline{0.973 \pm 0.03}$ | $\underline{0.889 \pm 0.20}$ | $\underline{0.981 \pm 0.01}$ | $\underline{1.123 \pm 0.28}$ | $\underline{0.980 \pm 0.01}$ | $\underline{1.074 \pm 0.51}$ | $\mathbf{0.972 \pm 0.02}$ | $\mathbf{1.165 \pm 0.50}$ | $\mathbf{0.969 \pm 0.02}$ |

Table 9: Reconstruction metrics across tokenizers for full backbone reconstruction with errors.

## K  EXTENDED THOUGHTS

This section contains extended discussion of several non-critical points raised in the main text. We welcome further discussion on any of the following.

### K.1  CONNECTIONS BETWEEN FLOW MODELS AND THE STRUCTURE DECODER

On first pass, it is somewhat surprising that flow autoencoders are so parameter efficient. The original structure module in Jumper et al. (2021) actually *reused* weights for each iteration of the structure decoder. This required careful tuning and several techniques to prevent instability during training. Subsequent methods that use the structure decoder in a generative or tokenization context generally opt not to share weights (Gaujac et al., 2024; Yim et al., 2023b). The structure decoder can be thought of as a very coarse diffusion model with only $8$ steps, where the prior is the identity rotation. Our transformer generalizes this to take an arbitrary number of steps. From this perspective, the performance of diffusion transformers (both in our work and in models like AlphaFold3) makes much more sense.

### K.2  WHY CAN POINTWISE LAYERS RECONSTRUCT SO WELL?

We found it intriguing that even very small window sizes with sliding window attention (SWA) gave good reconstructions, so we ablated a simple pointwise MLP. These ablations are discussed in Section 4.3. While these models underperformed our models with SWA, they nonetheless gave reconstructions of around $1.3$ Å. However, generative models trained over these codebooks performed very poorly. Why is this?

The purpose of the encoder in an image tokenizer like VQGAN is to pool local information to create meaningful representations. This has both a semantic and a technical motivation. Semantically, the encoder learns local components of an image, and powerful attention mechanisms can then learn long-range interactions. Technically, the downsampling reduces the sequence length of the transformer to a manageable size

Because our protein structure tokenizers operate on raw coordinates, there is already some long-range information present, since (roughly speaking) proteins near the start or the end of the chain

will often have bigger numerical values than ones near the center. Since this information is already present, the model just needs to learn to transform it in a way that's more amenable for sequence modeling, which is not that hard. Because generative models explicitly operate on *sequences* of tokens, however, this representation is not necessarily amenable to generation.

### K.3    THE CASE FOR AUTOREGRESSIVE GENERATION

Most models in the protein generation space are diffusion or discrete diffusion models, with good reason. These models work really well. They have saturated designability over the past year and they're proven to create proteins that can be designed in a lab. One could be forgiven for thinking that the focus on autoregressive models is unnecessary.

However, a major benefit of autoregressive models is that sequence lengths can be learned. The use of diffusion models seems to be heavily entrenched into the protein folding/inverse folding problem, **where the sequence length is known a priori**. When a model folds a sequence into a structure, by design it knows the structure size. Being able to generate protein sequences of variable lengths is a critical skill as we move towards more diverse tasks. The canonical example is *in situ* modeling, where one cannot purify a single protein so the sequence length is unknown. In principle, one could sample a bunch of different lengths with a diffusion model and score them, but this comes with its own set of issues. A human operator would need to make this decision beforehand and results would likely be quite brittle (e.g., the difference between having two and three proteins at inference time would be substantial). An autoregressive model could just learn to generate proteins of appropriate size a priori.

There are other examples (e.g., binding to membrane proteins, which is particularly important for therapeutics). With enough a priori information, a diffusion model generally outperforms an autoregressive model because it has full bidirectional communication. This gives diffusion models a significant advantage in benchmarks like designability, where people typically just generate proteins of different sizes. However, we generally think that as we move to more complex, in-the-wild tasks, giving our models the capability to reason about protein size will be a valuable capability.

