# OpenReview forum: "Flow Autoencoders are Effective Protein Tokenizers"
_ICLR.cc/2026/Conference — ICLR 2026 Poster_

### Official Review · Reviewer_BvV7 · 2025-10-25

**Soundness:** 4
**Presentation:** 4
**Contribution:** 2
**Rating:** 6
**Confidence:** 3

**Summary:**

They build a discrete VQVAE based tokenizer for protein structures, where the encoder maps 3D coordinates to discrete codes and the decoder is a DiT-based flow matching model. They additionally train an autoregressive prior over the latent codes for generation. The approach, Kanzi, simplifies aspects of prior approaches, including the use of a diffusion loss (as opposed to prior SVD-based losses), replacing frame representations with global coordinates, and using standard attention instead of SE(3) invariant architectures. Kanzi outperforms prior token-based structure generative models and is more parameter efficient.

**Strengths:**

- The authors perform extensive benchmarking against prior approaches on both reconstruction and generative tasks. The results demonstrate that the model is competitive against the prior SOTAs on both tasks.
- They perform a series of ablations on encoder variants (invariant vs not invariant), attention window sizes, and model size.
- Simplifying the auto-encoder loss is pretty significant, as it significantly reduces the cost of training from O(L^3) or O(L^2) to O(L).
- Using a diffusion decoder also allows you to use classifier-free guidance and the diffusion noise scales to balance the tradeoff between diversity and sample quality.

**Weaknesses:**

- The idea isn't incredibly novel. The paper is a combination of many design choices (diffusion decoder, DiT rather than SE(3) invariant attention, FSQ discretization) rather than a single great idea.
- You mentioned the cost of loss functions used in prior works. Maybe you can do an experiment measuring the iteration speeds and memory scaling of each approach? Note also that diffusion model autoencoders have a multiplicity (number of diffusion timesteps per step), which can increase the memory use / runtime of this approach. Also, despite these supposed gains, you only trained on proteins of length <256. An experiment scaling to larger systems may be worthwhile.

**Questions:**

- Do you think the sliding window attention would still suffice for longer sequences?
- While the model achieves superior reconstruction and competitive generation for token‐based methods, it still lags continuous diffusion models. What are the main causes of this gap?

---

> ### Author Response · Authors · 2025-11-21
> **Response to reviewer BvV7**
>
> Thank you for acknowledging the strength and significance of our approach. We've addressed your questions / comments as follows.
>
> 1. **Scaling to longer sequences.** This is an interesting question, so we did an additional ablation with the same model size and window size up to protein size 512 (the 256 limit was mostly for computational reasons). The below table shows the results at this model size evaluated only on long proteins ($256\leq L < 512$). We still achieve excellent reconstructions. Despite the fact that proteins contain long range interactions, empirically it seems that sliding window attention suffices for larger systems, at least for reconstruction. It's also worth emphasizing that the long sequence performance with a relatively short window size is particularly encouraging for scaling to complexes or much longer proteins, since we do not need to actually instantiate the full attention matrix. We've added this ablation to the Appendix.
> 2. Note that Appendix I contains a description of FAPE vs Kabsch vs dRMSD. We do want to emphasize that the dominant cost is not in memory (the memory from the loss functions is quite small compared to the activations throughout the model trunk, particularly for these large models, and FAPE/dRMSD actually parallelize fairly well on GPUs), but rather complexity and scaling. Diffusion transformers have successfully scaled across domains. Models that use FAPE / dRMSD generally need several auxiliary losses to balance early training. They also have odd failure modes; in the Appendix, we show that the variance of ESM-3's performance is very high. Despite achieving excellent mean reconstruction, it fails catastrophically on a nontrivial subset of proteins, with RMSDs as high as 20A. We suspect this is why the only prior tokenized models that have demonstrated strong generative performance are DPLM-2 and ESM-3, which were both trained on massive amounts of data.
>
> 4. It's not obvious how to compare the approaches during training, since the model architectures will look quite different. During inference, our decoding is much slower since we need to integrate the full SDE (though recent distillation methods suggest this can be made much faster, there just hasn't been a significant need). Our encoder is quite shallow, so it tends to be faster than most other methods.
> 5. The point about multiplicity is true; we generally sample multiple diffusion timesteps as different elements in a batch.
>
>
> We appreciate the observation that many of the individual components of Kanzi have been explored earlier. Our contribution is not inventing a single new primitive, but rather exploring how a principled combination of these elements leads to high fidelity tokenization. We think many of these insights are non-obvious -- for example:
>
> 1. The observation that sliding window attention can reconstruct so well was unexpected and enables scaling to long proteins.
> 2. The observation that pointwise layers alone can reconstruct but not generate is an important design detail for building better tokenizers.
> 3. The weak performance of invariant representations and learned-invariant representations runs against the standard tokenization paradigm for proteins (which usually follows geometric/invariant point attention)
>
> We also make more engineering heavy contributions that, to our knowledge, have not appeared elsewhere (e.g., the use of an asymmetric encoder/decoder design, the use of shared adaLN layers).
>
> We think the most important finding, however, is the fact that we can match or outperform much larger, more complex models. Proteins are ultimately somewhat simple structures, so it's somewhat startling that all the tokenizers to date have needed expansive pretraining. Our work implies that much of this complexity can be removed; much like the manifold hypothesis in images, the actual protein distribution is quite low dimensional.
>
> **Discrete vs continuous:** The gap between discrete token based methods and continuous diffusion models is a very good question. This is not just the case for proteins; diffusion models are state-of-the-art in computer vision (though recent approaches have challenged this dominance). While we are still exploring this question, we think one major factor is proteins are very sensitive to small changes in atomic coordinates. Gaussian noise of ~0.2 A is sufficient to substantially distort ProteinMPNN results. A diffusion model can make small adjustments in the last few steps of inference to account for this. In tokenized models (whether autoregressive or discrete diffusion), once a token is sampled it is fixed. Errors exposure during sampling thus has to be "recovered" by the decoder, which is quite challenging.
>
> ### Long sequence performance (256 <= L < 512)
> | Dataset | Reconstruction (A) |
> |---------|--------|
> | cath    | 1.044 |
> | casp14  | 0.835 |
> | casp15  | 0.909 |
> | cameo   | 0.898 |
> | afdb    | 1.051 |

---

### Official Review · Reviewer_zKJZ · 2025-10-29

**Soundness:** 2
**Presentation:** 3
**Contribution:** 2
**Rating:** 4
**Confidence:** 4

**Summary:**

This paper introduces Kanzi, a non–SE(3)–equivariant diffusion autoencoder–based protein structure tokenizer, trained using a single flow-matching objective on global 3D coordinates. The authors argue this architecture simplifies training relative to SE(3)-invariant tokenizers and demonstrate strong reconstruction and unconditional structure generation performance compared to recent tokenizers such as ESM3, DPLM2, FoldToken, and IST.

**Strengths:**

- Clear motivation to simplify protein structure tokenization by removing architectural and loss engineering complexity.

- Technically solid implementation using flow-matching and FSQ-based codebooks.

- Strong reconstruction quality across multiple protein benchmarks despite relatively small model size and training data.

- Interesting empirical findings: non-equivariant encoders outperform equivariant ones under flow objectives, codebook utilization emerges late in training.

- Introduction of rFPSD as a distributional reconstruction metric is potentially valuable.

**Weaknesses:**

1. No evaluation of SE(3) stability / invariance — critically, the tokenizer may output different tokens for rotated versions of the same protein, which invalidates many downstream use cases (retrieval, clustering, homology, interpretability). This is not even measured.

2. No retrieval / similarity / StructTokenBench[1]-style evaluation, even though retrieval is a core purpose of structured tokenization (cf. FoldSeek[2], ).

3. No conditional generation experiments, despite stating generative capability as a core contribution; all reported results are unconditional only.

4. Claims of “smaller model rivaling ESM3/DPLM2” are not apples-to-apples — those are protein sequence / multimodal models, not structure-only.

5. Scope of “tokenizer quality” is too narrow — heavily focused on reconstruction, insufficient multi-dimensional evaluation (e.g. sensitivity, explainability, stability, controllability).

[1] Protein Structure Tokenization: Benchmarking and New Recipe

[2] Fast and accurate protein structure search with Foldseek

**Questions:**

1. Does rotating a protein structure change the tokenization output? Have you evaluated rotational consistency quantitatively?

2. Why is retrieval / homology search omitted? Do Kanzi tokens perform poorly under similarity search?

3. Can Kanzi support conditional generation (e.g., topology, motif, scaffold constraints)? If yes, why is it not reported?

4. You do not perform residue-level local centering or relative-frame coordinate normalization (e.g., per-residue local frame / backbone-centric coordinates) — without such normalization, how are the learned tokens supposed to be interpretable or physically meaningful?

---

> ### Author Response · Authors · 2025-11-21
> **Response to reviewer zKJZ**
>
> Thank you for your detailed feedback and critiques. We’ve sought to address your main concerns below.
>
> 1. The tokenizer will generally output different tokens for rotated versions of the same protein. This is a design consideration, and we demonstrate a specific instance (cryoET docking) where this is actually desirable. It isn't obvious how to disentangle the pose from the structural encoding; however we do include an ablation (in Appendix G) where we attempt to directly learn an invariant representation. The tradeoff in ease of training and performance is not worthwhile. Note that we believe most of the applications for which invariance is valuable can also be achieved with non-invariant tokenizers, see point (2).
> 2.  Evaluations of Kanzi’s performance on non reconstructive/generative tasks is an excellent recommendation. Based on StructTokenBench, we have added evaluations demonstrating that our tokenizer learns useful representations, despite the lack of invariance. We focus on residue level properties (catalytic site prediction, conserved site prediction, repeat motif prediction, epitope prediction, binding site prediction, and structural flexibility), rather than global properties. We find that Kanzi tokens outperform other non-invariant tokenizers like bio2token but largely underperform invariant tokenizers like ESM-3 or DPLM-2. See the table below for more details. In retrospect, this is not particularly surprising, since most residue probing experiments are fundamentally local in nature (e.g., detecting if a site is a binding site). Probing experiments on non-invariant tokenizers mean a single MLP must learn to transform global representations to local ones, which is quite challenging.
> 3. The suggestion to include conditional generation is a valuable addition. To support the claim that Kanzi can support conditional generation, we have included results showing how Kanzi tokens enable a cryoET language model (see Appendix D for more details). The cryoET docking problem seeks to find a protein structure matching a 3D volumetric image of a macromolecular *in situ* complex. This is different in kind from the cryoEM problem (where the protein sequence is known, so one can employ tools like AlphaFold) and the denoising problem (where the goal is to go from noisy views to a clean tomogram). We demonstrate an initial proof-of-principle experiment that we can reconstruct protein structures from cryoET data to ~1-3 A precision using a language model trained on Kanzi tokens. We emphasize that this is a **new capability** because invariant tokens struggle with non-invariant conditioning, and the volumetric conditioning is intrinsically non-invariant. We can also demonstrate reasonably strong results by conditioning on CATH code (as was originally done in Proteina); we are happy to include these results as well if desired.
> 5. Our primary intention with the comparisons to ESM3 and DPLM2 was to focus on the structure component, since structure tokenizers are a critical component of building multimodal models. We have adjusted the language in the paper to emphasize that our comparison is strictly against the respective structure tokenizers. Note that the parameter and data counts described in the paper already only reflect the structure encoder/decoders. The ESM3 structure decoder alone has >600M parameters, and the DPLM2 structure encoder is a pretrained GVP-Transformer trained on millions of structures with >100M parameters.
> 6. As mentioned in (2), we have expanded the tokenizer quality experiments to include residue-level probing across six different experiments along with codebook analyses (as described in [1]).
>
> Thank you for your feedback; we believe the additions were very valuable for improving the quality of the paper. Please let us know if there are further extensions and ablations you would like to see added.
>
> | Model     |   bindint    |               |   catint     |               |   con        |               |   ept        |               |   rep        |               |   phys       |               |
> |-----------|:------------:|:-------------:|:-------------:|:-------------:|:-------------:|:-------------:|:-------------:|:-------------:|:-------------:|:-------------:|:-------------:|:-------------:|
> |           | Fold | Family | Fold | Family | Fold | Family | Fold | Family | Fold | Family | Fold | Family |
> | bio2token | 0.489 | 0.637 | 0.544 | 0.539 | 0.512 | 0.543 | 0.514 | 0.525 | 0.503 | 0.569 | 0.312 | 0.268 |
> | dplm2     | 0.540 | 0.794 | 0.598 | 0.703 | 0.570 | 0.728 | 0.623 | 0.700 | 0.507 | 0.763 | 0.475 | 0.433 |
> | esm3      | 0.497 | 0.787 | 0.549 | 0.749 | 0.541 | 0.647 | 0.614 | 0.644 | 0.519 | 0.677 | 0.433 | 0.414 |
> | kanzi     | 0.504 | 0.692 | 0.531     | 0.603     | 0.525 | 0.626 | 0.532 | 0.604 | 0.526 | 0.614 | 0.298 | 0.357 |
>
> [1] Yuan, Xinyu, et al. "Protein structure tokenization: Benchmarking and new recipe." arXiv preprint arXiv:2503.00089 (2025).

---

### Official Review · Reviewer_xSwq · 2025-11-03

**Soundness:** 3
**Presentation:** 3
**Contribution:** 3
**Rating:** 6
**Confidence:** 3

**Summary:**

This paper presents Kanzi, a flow-based protein structure tokenizer that replaces SE(3)-invariant components with a simpler diffusion autoencoder trained using a flow matching loss. The approach removes the need for complex invariant losses and geometric attention while maintaining or even improving reconstruction performance compared to prior tokenizers like ESM3, DPLM2, and FoldToken.

Kanzi is then used to train an autoregressive model (Kanzi-AR) for structure generation, demonstrating competitive designability and diversity.

**Strengths:**

- **The experimental setup is solid and shows careful design choices** — e.g., diverse test datasets, detailed RMSD/TM metrics, and fair consideration of computational trade-offs.

- **The writing is clear, and figures are well-designed.** The methodology section carefully explains the training and inference setup.

- **Simplified yet effective formulation.** The use of a single flow matching loss instead of multiple SE(3)-invariant losses reduces training complexity and improves stability. This work is more like an altogether engineering refinement than a conceptual breakthrough. The proposed simplification aligns with recent diffusion autoencoder trends in vision. Though the novelty in biological context is limited, the "making it simple and scalable" is indeed a very important next step for this field.

**Weaknesses:**

- **Performance is not that out-standing**: The reconstruction performance and generation performance do not look that out-standing. ESM3 seems to still be the best of all. Inference-time sampling tricks from Proteina seems to be able to boost Kanzi-AR to a next level, though unfortunately neither ESM3 or DPLM2 used this trick. This might be an unfair comparison.

- **Only evaluating on reconstruction and generation, missing representation quality**: From image domain, there are some papers discussed about one point: not necessarily the best reconstruction quality leads to a better tokenizer. The representation quality also matters. e.g., see the table 1 in RAE paper (https://arxiv.org/pdf/2510.11690v1) for MAE-B and DINOv2-B. And there is a benchmark designed for structure tokenizer representation quality: from AminoAseed [1] paper. Highly suggest to also benchmark representation quality rather than reconstruction quality alone.

- **Missing comparisons to other structure tokenizer benchmarks**: see questions.

- **Minor typos**: There are also minor typographical issues (e.g., lines 794–797 contain “¿” and “¡” characters).


[1] Protein Structure Tokenization: Benchmarking and New Recipe

**Questions:**

1. Compare with two more baselines:
- Cheap [1]
- AminoAseed [2]

2. Add representation quality evaluation. For example, the benchmark from AminiAseed [2].

3. Questionable use of diffusion sampling in AR models. Table 3 claims Kanzi-AR uses inference-time sampling tricks (Eqn. 5), which conceptually apply to diffusion-based samplers, not discrete autoregressive decoders. As I can understand, AR simply uses the discrete structure tokens for autoregressive training and sampling. There should not be any diffusion sampling process involved? Clarification is needed.

[1] Tokenized and continuous embedding compressions of protein sequence and structure

[2] Protein Structure Tokenization: Benchmarking and New Recipe

---

> ### Author Response · Authors · 2025-11-21
> **Response to reviewer xSwq**
>
> We sincerely appreciate your acknowledgment of the simplicity and scalability of our approach. We've addressed some of your concerns below.
>
> 1. Re performance, Table 1 shows that Kanzi generally trades off best vs second best performance with ESM3. Even in the cases where Kanzi is second-best, the difference is generally on the order of 0.1 A. This is despite the fact that **Kanzi was trained on much less data and is much smaller than ESM-3**.
> 2. An important under-discussed point is that on secondary structure such as beta sheets and coils, Kanzi consistently outperforms all other methods, including ESM3. This is shown in Table 1. Structued alpha helices are much simpler to reconstruct than the more disordered beta sheets and coils. As many biological applications involve non-structured data (e.g., designing flexible CDR regions for antibodies), we think this capability is a significant comparative improvement.
> 3. Out of the box, ESM-3 is not particularly strong in terms of designability. We trained a separate model (ESM3-AR) that improves on this slightly by using more sophisticated sampling, but providing the same sampling methods to Kanzi-AR shows that Kanzi-AR outperforms ESM3-AR on structural generation.
> 4. We  do wish to clarify that there are several inference time sampling tricks at play. *Noise/score annealing* is not a trick developed by Proteina, but has rather been a part of the protein diffusion literature for quite some time. These methods, however, are not applicable to discrete methods with single pass decoders like ESM-3 and DPLM-2, since there is no closed form score or velocity field as there is for continuous methods like Kanzi and Proteina. *Best-of-N sampling*, which we use with Kanzi-AR is a more general technique (also not from Proteina). It is explicitly incompatible with ESM3/DPLM2, as those are not autoregressive models and there isn’t a closed form likelihood as a reward proxy for best-of-N. We didn't see much effect when using best-of-N with DPLM and ESM3-AR. Thus, we think the comparisons are still robust -- please let us know if there are any confusing points here!
> 5. This is an excellent point; we've since added representation experiments based on the AminoAseed benchmark to the Appendix. As a high level summary, Kanzi tokens generally underperform invariant representations on residue level probing tasks. We find this unsurprising; most of these tasks rely on local neighborhood information moreso than global information, and shifting to a local reference frame is a challenging operation for a simple probing MLP to learn. Please see the updated Appendix E for more information!
> 6. Re AminoAseed, we were unable to get positive reconstructions using the released model weights, nor were we able to reproduce the paper results from the repo online. We will update this section once we're able to get reproducible results. CHEAP is a joint-sequence structure tokenizer, so many of the metrics we care about (e.g., reconstruction, rFPSD) aren't quite applicable (in particular, CHEAP operates in the ESM latent space, so a reconstruction experiment is sort of checking how well it preserves the ESMFold latent space). Is there a particular comparison you'd like to see?
> 7. To clarify the diffusion approach, the autoregressive model generates a sequence of tokens; these tokens condition a diffusion decoder. Thus, the inference-time sampling tricks are used *during decoding*. You are correct that during generation there is no diffusion sampling. We've changed the paper language to clarify this point.
>
>
> Thank you again for all your detailed feedback! Please let us know if there are any further clarifications needed.

---

### Official Review · Reviewer_6hfk · 2025-11-03

**Soundness:** 4
**Presentation:** 3
**Contribution:** 4
**Rating:** 10
**Confidence:** 4

**Summary:**

The authors present a flow-based protein structure auto-encoder ("Kanzi") useful as a protein tokenizer. The primary contribution is that, as far as I am aware, it is the only example of flow-based structure auto-encoder. Secondary to that, but still very important, is that the encoder and decoder are not invariant or equivariant, continuing the recent trend to show that such complex models are not required to adequately model proteins. The authors additionally use the model to generate novel structures.

**Strengths:**

The authors presentation is mostly very clear, and it does a good job of discussing the previous literature. They present a novel approach to the problem of structure tokenization. I particularly encouraged to see that the output is simply the backbone atomic coordinates, which should greatly simplify the tokenizer's use. It is intriguing that the model uses real space coordinates _and_ a sequence id embedding; see questions for more on this.

I like that the authors show that the resulting tokenizer can be used to generate designable structures. But see weaknesses for additional comments on this.

Clearly Kanzi is successful: the 30M parameter version trained with "under optimized" hyperparameters is best or second best across most metrics, covering the core set of structure prediction data, while being _much_ smaller.

The discussion of ablations is especially useful, although I do wish it were more detailed.

**Weaknesses:**

This is a small weakness but the authors seem to flip between using "diffusion" and "flow matching" to describe their approach. It seems that flow matching better matches what they're doing--either that or there is something that needs to be clarified substantially in their writing. In any case diffusion and flow matching are not exactly the same thing, so the authors should be precise.

There are two significant weaknesses in the paper that I'd like to see addressed:
1. The authors should examine how their tokenizer performs at other downstream tasks that are relevant for protein language models. Many are detailed in a recent paper from ICML 2025: Xinyu Yuan et al., Protein structure tokenization: Benchmarking and new recipe.

2. While the authors show that their auto-regressive structure generator is capable of generating designable structures using the conventional definition of generating a sequence which folds back to the same structure, this is highly dependent on the particular folding model used. The authors chose ESMFold, which I found curious given they train on a sample of AFDB. The authors should comment on this choice given that better models are available.

**Questions:**

The authors might want to cite Ellmen, ... Deane "Transformers trained on proteins can learn to attend to Euclidean distance", which includes an interesting discussion of how attention mechanisms process real space coordinates.

---

> ### Author Response · Authors · 2025-11-21
> **Response to reviewer 6hfk**
>
> Thank you for acknowledging the value of our contribution! In response to your comments, we'd like to highlight the following changes/additions we made.
>
> 1. Changed the paper to consistently use "flow matching" over diffusion. We maintain the use of "diffusion" in a few places (e.g., when describing other work). We also added a citation to (Ellmen), which we think is quite relevant in light of our finding that pointwise layers can reconstruct but not generate.
> 2. This is an excellent point; we've added a table to the appendix showing the performance of Kanzi tokens on residue level benchmarking tasks based on StructTokenBench. As a high level summary, Kanzi tokens generally outperform other non-invariant methods but underperform invariant representations on residue probing tasks. We find this unsurprising; most of these tasks rely on local neighborhood information moreso than global information, and shifting to a local reference frame is a challenging operation for a simple probing MLP to learn.
> 3. We also added a conditional generation benchmark, where we demonstrate that Kanzi tokens can be used to reconstruct proteins from cryoET data on a synthetic dataset. This is a new capability that the non-invariant approach unlocks, because invariant tokens generally struggle to use non-invariant conditioning (which is intrinsic to the cryoET problem). See Appendix D for more details.
> 4. We mostly chose ESMFold because (1) it is fairly standard in the literature, see [3-5] and others, and (2) it is reasonably fast and performant. While models like AlphaFold3 are probably more accurate, they also require computing an MSA, which is computationally expensive, particularly when one wishes to obtain statistics over a large number of structures. The primary alternative would be OmegaFold. To our knowledge designability calculations don't tend to change that much across ESMFold and OmegaFold, but if this is a concern we're happy to ablate it.
>
> [1] Sargent, Kyle, et al. "Flow to the mode: Mode-seeking diffusion autoencoders for state-of-the-art image tokenization." arXiv preprint arXiv:2503.11056 (2025).
>
> [2] Chen, Yinbo, et al. "Diffusion autoencoders are scalable image tokenizers." arXiv preprint arXiv:2501.18593 (2025).
>
> [3] Geffner, Tomas, et al. "Proteina: Scaling flow-based protein structure generative models." arXiv preprint arXiv:2503.00710 (2025).
>
> [4] Bose, Avishek Joey, et al. "Se (3)-stochastic flow matching for protein backbone generation." arXiv preprint arXiv:2310.02391 (2023).
>
> [5] Lin, Yeqing, et al. "Out of many, one: Designing and scaffolding proteins at the scale of the structural universe with genie 2." arXiv preprint arXiv:2405.15489 (2024).

---

### Author Response · Authors · 2025-12-03
**Global Response**

We thank all reviewers for their feedback and overall positive reviews, and the AC for taking the time to review this work. The reviewers highlighted the strong performance of our work, strength of our experimental design, and clarity of our presentation. We responded to all concerns in individual replies to the reviewers. Here, we summarize (briefly) our main contributions and additional changes/experiments we made in response to reviewer concerns.

This work introduced a novel approach to biomolecular tokenization, where the decoder is a diffusion tokenizer. In line with recent work, our we use simple, scalable architectures over SE(3)-invariant components. The resulting tokenizer significantly outperforms almost all other biological tokenizers; it matches the performance of ESM-3, which is trained on 400x as much data and is 20x larger by parameter count. As noted by reviewers, ours is the first work to train a protein tokenizer using a diffusion/flow matching approach (6hfk) and substantially simplifies the tokenization training process (xSwq).

**Representation quality.** A shared concern by several reviewers is the lack of representation tasks. We have since added a more detailed section to the paper Appendix demonstrating our tokenizers downstream performance on six different residue-level representation tasks. See Table 4 in Appendix E for numerical results and discussion.

**Long sequence performance.** Reviewer BvV7 asked whether the use of sliding window attention would suffice for longer sequences, as our base models are trained on sequences up to 256 residues. We performed an additional ablation demonstrating that sliding window attention with window size 8 is sufficient for strong reconstruction performance on longer sequences between 256 and 512 residues. See Table 6 in Appendix G for numerical results.

**Conditional generation.** Reviewer zKJZ commented on the lack of conditional generation baselines. While our approach is compatible with most conditional tasks, we include results showing how Kanzi tokens enable a cryoET language model (see Appendix D). This model answers the task of generating a protein structure fitting a particular cryoET image. We emphasized this task as it is a new capability enabled by our non-invariant approach; invariant tokens generally struggle to condition non-invariant data and vice versa.

We believe these additional results significantly strengthen our paper, and again thank the reviewers for their valuable suggestions.

---

### Meta-Review · Area_Chair_iAcH · 2026-01-24

**Summary:**

The main concerns about the papers include
1. Performance and comparison with baselines (by Reviewer xSwq, zKJZ, and BvV7)
2. Lack of physical interpretability and questions about SE(3) invariance (by Reviewer zKJZ)
3. Efficiency and scalability (Reviewer BvV7)

**Reviewer Concerns:**

Authors provided additional experiments, including additional baseline and benchmarks, as well as many clarifications related to experiments and design choices (such as those about SE(3) invariance). Most of the concerns are at least partially addressed.

**Reviewer Scores:**

I beleive reviewer 6hfk, xSwq, and BvV7 will remain positive after the rebuttal. Reviewer zKJZ might also lean more positive after reading the rebuttal.

---

### Decision · Program_Chairs · 2026-01-26

Accept (Poster)